# Challenges in Simulating Ozone Depletion Events in the Arctic Boundary Layer: A Case Study Using ECHAM/MESSy for Spring 2019/20

Stefanie Falk<sup>1</sup>, Luca Reißig<sup>1</sup>, Bianca Zilker<sup>2</sup>, Andreas Richter<sup>2</sup>, and Björn-Martin Sinnhuber<sup>1</sup>

**Correspondence:** Stefanie Falk (stefanie.falk@kit.edu)

**Abstract.** Ozone depletion events (ODEs) and bromine explosions (BEs) occur regularly in the springtime polar boundary layer. ODEs alter the oxidation capacity of the polar boundary layer and promote formation of toxic mercury. We investigated Arctic ODEs and BEs in 2019/20 using the chemistry-climate model ECHAM/MESSy v2.55.2, nudged with ERA5 reanalysis data. Model results were evaluated against surface ozone measurements, satellite-derived tropospheric BrO vertical column densities (VCDs), and in situ data from the MOSAiC expedition. The model underestimated boundary layer (BL) height during shallow BL conditions, coinciding with a warm surface temperature bias  $(2-10\,\mathrm{K})$ , particularly below  $-10\,^\circ\mathrm{C}$ , likely inherited from ERA5. An updated model configuration, incorporating more realistic multi-year sea ice and relaxed bromine release thresholds, improved agreement with coastal ozone observations (Eureka, Utqiagvik) but still failed to reproduce strong ODEs observed during MOSAiC. Consequently, modeled surface BrO mixing ratios were overestimated, while BrO VCDs were underestimated, suggesting that simply increasing Br<sub>2</sub> emissions does not resolve discrepancies. A weaker colocation between modeled BrO VCDs and ODEs aligns with prior airborne studies and may reflect tropospheric chemical and transport processes rather than stratospheric contributions. Despite decreasing Arctic sea ice extent and increasing BrO VCDs, long-term records from Alert, Utqiagvik, and Zeppelin show a decline in strong ODE frequency since 2008. This suggests that bromine emissions from first-year sea ice alone may not fully account for observed ODE variability, and that additional climate-sensitive mechanisms may modulate Arctic ozone chemistry. Long-term model integrations are recommended to better understand these trends.

#### 1 Introduction

The Arctic is subject to rapid changes due to the warming climate (AMAP, 2012; Rantanen et al., 2022). The impact of this change is reflected by the decrease of sea ice extent and thickness over the past decades (Stroeve and Notz, 2018; Lindsay and Schweiger, 2015) which caused a decline in multi-year sea ice and a stronger seasonality of the ice cover (Haine and Martin, 2017). Sea ice extent and thickness are projected to further decrease in all future projections (Notz and SIMIP Community, 2020) which may open new routes for cargo ships towards the end of the 21st century with an associated increase in Arctic

<sup>&</sup>lt;sup>1</sup>Institute of Meteorology and Climate Research, Atmospheric Trace Gases and Remote Sensing, Karlsruhe Institute of Technology, Karlsruhe, Germany

<sup>&</sup>lt;sup>2</sup>Institute of Environmental Physics, University of Bremen, Bremen, Germany

air pollution (Yumashev et al., 2017). The composition of the Arctic troposphere in winter and spring is strongly affected by long-range transport from mid-latitudes, while the polar dome prevents this transport in summer (Bozem et al., 2019). Primary pollutants include  $NO_x$  and CO, which are precursor substances for the formation of tropospheric ozone  $(O_3)$  after polar sunrise. Annual mean ozone volume mixing ratios (VMRs) at the surface are relatively low in the Arctic ( $\chi_{O_3} \leq 30 \,\mathrm{ppbv}$ ) compared to the mid-latitudes of the northern hemisphere and show a distinct seasonal cycle with a polar winter maximum (October–February) and a polar summer minimum (March/April–August/September) (Helmig et al., 2007; Whaley et al., 2023).

Tropospheric ozone VMR has been observed to drop below the detection limit on sites near the Arctic Ocean in springtime (March–May) for decades. These periods of  $\chi_{\rm O_3} \leq 5\,\mathrm{ppbv}$ , sometimes extending for several days, were first recognized by Bottenheim et al. (1986) in Alert (Canada), who coined the term Ozone Depletion Events (ODEs). Later, satellite observations revealed coincident plumes of enhanced bromine monoxide (BrO) vertical column densities (VCDs) extending over synoptic scales in both Arctic and Antarctic regions (Richter et al., 1998; Wagner and Platt, 1998; Richter et al., 2002). This was evidence that bromine (Br) chemistry leads to the destruction of ozone during ODEs (Barrie et al., 1988; McConnell et al., 1992; Platt and Hönninger, 2003). The associated chemical mechanism also alters the oxidation capacity of the polar boundary layer and affects formation of toxic, oxidized mercury (Brooks et al., 2006; Wang et al., 2019c) that accumulates up the food chain and causes serious health issues.

30

With polar sunrise, molecular bromine  $(Br_2)$  is photo-dissociated (Eq. (R1)) and subsequently destroys  $O_3$  forming BrO 40 (Eq. (R2)):

$$Br_2 + h\nu \rightarrow 2Br,$$
 (R1)

$$Br + O_3 \rightarrow BrO + O_2$$
. (R2)

The resulting BrO can then self-react to form both Br and Br<sub>2</sub> (Eq. (R3)) or react with hydroperoxyl (HO<sub>2</sub>) to form hypobromous acid (HOBr) (Eq. (R4)), which photo-dissociates to Br (Eq. (R5)) starting the cycle again:

45 BrO + BrO 
$$\rightarrow$$
 
$$\begin{cases} 2Br + O_2 \\ Br_2 + O_2 \end{cases}$$
, (R3)

$$BrO + HO_2 \rightarrow HOBr + O_2,$$
 (R4)

$$HOBr + h\nu \rightarrow OH + Br.$$
 (R5)

These reactions terminate with the formation of reservoir species such as hydrogen bromide (HBr) that are not efficient at forming reactive bromine. The main source of inorganic bromine in the polar regions is bromide (Br $^-$ ) from sea salt (Koop et al., 2000). It is continually emitted from the open ocean or open leads in the sea ice. Sea salt aerosols release Br $_2$  into the gas-phase (Yang et al., 2005). A minor contribution comes from organic bromine (for example CHBr $_3$ , CH $_2$ Br $_2$ , CH $_3$ Br) emitted from ocean and sea ice (Stemmler et al., 2015; Abrahamsson et al., 2018). Recycling of inorganic bromine is necessary to sustain ODEs (Abbatt et al., 2012). This can involve heterogeneous and multiphase reactions of, for example, HOBr or bromine nitrate (BrNO $_3$ ) on salty ice surfaces or in salty solutions (Sander et al., 2006):

55 
$$HOBr + \begin{cases} Br_{aq}^{-} & \xrightarrow{H^{+}} \begin{cases} Br_{2} \\ BrCl \end{cases} + H_{2}O,$$
 (R6)

$$BrNO_3 + \begin{cases} Br_{aq}^- \\ Cl_{aq}^- \end{cases} \rightarrow \begin{cases} Br_2 \\ BrCl \end{cases} + NO_3^-.$$
(R7)

Interhalogen reactions are able to turn BrCl into Br<sub>2</sub>:

$$BrCl + Br^- \rightleftharpoons Br_2Cl^- \rightleftharpoons Br_2 + Cl^-.$$
 (R8)

For each brominated trace gas in Eqs. (R6)–(R7) getting in contact with an icy surface of high salinity, twice the number of bromine atoms is released. This exponential increase is called bromine explosion (BE).

This mechanism is well understood through lab experiments (Fickert et al., 1999; Adams et al., 2002) and physicochemical aspects of bromide oxidation on salty ice surfaces including the role of acidity and temperature have been studied in detail (Wren et al., 2010; Oldridge and Abbatt, 2011). Box modeling (Sander et al., 2006) and one-dimensional model studies that also includes deeper snow layers (Toyota et al., 2014) have shown that the picture is far from complete and that liquid-phase reactions also play an important role in deeper snow layers. Accurate predictions on a larger scale still pose a challenge for Chemistry Transport (CTMs) and Chemistry-Climate Models (CCMs) in their default setup (Whaley et al., 2023).

Observational evidence indicates that BEs occur in two distinct synoptic situations: stable boundary layer (BL) with little mixing, and turbulent mixing during blizzards (Hausmann and Platt, 1994; Jones et al., 2009; Zhao et al., 2016). This is reflected by two distinct mechanisms for bromine release implemented into models: (1) surface bulk snow and ice (Toyota et al., 2011) and (2) blowing snow emissions (Yang et al., 2008, 2010). Marelle et al. (2021) showed that bromine emissions from bulk snow and ice dominate over those from blowing snow in triggering ODEs while blowing snow is an important additional source of sea salt aerosols.

Latest satellite observations of BrO VCDs with a high spatial resolution (Seo et al., 2019, 2020) indicate that snow-covered first-year sea ice in combination with open leads and sea ice cracks may act as the primary source of bromide during BEs which is in contrast to earlier findings that showed no coincidence between these sea ice features and the intensity of ozone depletion (Ridley et al., 2003).

This work aims at a better understanding and characterization of the bromine release from sea ice and surface snow and the subsequent BEs and ODEs in the global CCM ECHAM/MESSy Atmospheric Chemistry (EMAC) model (Jöckel et al., 2005, 2016). To assess the model's performance in the central Arctic, we utilize observational data from the 2019/20 season, corresponding to the Multidisciplinary drifting Observatory for the Study of Arctic Climate (MOSAiC) expedition. In Sect. 2, we describe the EMAC model and present the observational data sets used for model evaluation. The observational data consist of O<sub>3</sub> VMRs at Arctic ozone monitoring sites and satellite BrO VCD at high resolution from the TROPOMI instrument onboard the Copernicus Sentinel-5P satellite. Subsequently, we identify the best model setup based on spring 2019 conditions (Sect. 3). Results using this best setup are compared to observational data for spring 2020 especially accounting for measurements taken during the MOSAiC expedition. Implications on pan-Arctic and site-specific climatological ozone depletion trends as well as BrO VCDs are presented in Sect. 4. Finally, we discuss (Sect. 5) and summarize our results giving a brief outlook in Sect. 6.

### 2 Methods and data

#### 2.1 Model description and setup

We use the Modular Earth Submodel System (MESSy) v2.55.2 in combination with basemodel ECHAM5.2 (ECHAM/MESSy) referred to as EMAC. MESSy is a software framework that combines Earth system components, such as atmosphere, land, ocean, and subsystems of these, in a modular way (Jöckel et al., 2005, 2016). This modularity enables the deployment of different basemodels with an identical numerical implementation of geophysical and -chemical processes and parameterization.

We base the EMAC model setup on REF-C1SD of the Chemistry Climate Model Initiative (CCMI) 2 (Orbe et al., 2020). We use specified dynamics (SD) for the historical period 2019/20, with nudged surface pressure (logarithmic) and nudged temperature, divergence, and vorticity fields from above the boundary layer (model level 8) up to a pressure level of 1hPa. Sea Surface Temperatures (SSTs) and Sea Ice Cover (SIC) are also prescribed. Nudging data were generated from the European Centre for Medium-Range Weather Forecasts (ECMWF) Reanalysis v5 (ERA5) (Hersbach et al., 2020).

All simulations were conducted in a T42L90MA resolution with the spectral truncation T42 that translates to a  $(2.7851 \times 2.8125)^{\circ}$  regular Gaussian grid in the polar regions (latitude  $> \pm 68^{\circ}$  in both hemispheres). Our model setup comprises 90 hybrid pressure levels up to  $0.01\,\mathrm{hPa}$ . These hybrid pressure levels are terrain following, resulting in about 1–3 levels within the Arctic spring-time BL. In the Arctic, these levels translate on average to heights above ground at the upper boundary of each level as follows: level:  $(1,2,3,...,8) \rightarrow \mathrm{height}: (99,254,538,...,3571)\,\mathrm{m}$ .

We use the Module Efficiently Calculating the Chemistry of the Atmosphere (MECCA) (Sander et al., 2019) with the CCMI-base-02 chemistry mechanism in which mercury and iodine chemistry are switched off. This chemical mechanism comprises 261 gas-phase, 12 heterogeneous, and 80 photolysis reactions (Supplement A). The explicit calculation of heterogeneous reactions is restricted to the stratosphere and uses a prescribed aerosol surface concentration climatology (1996–2005) computed with the MESSy submodel MADE (Ackermann et al., 1998). In the troposphere and boundary layer, we parameterize the

Figure 1. Schematic of bromine explosions (BEs) and catalytic ozone depletion reaction network separated by families. The emission of  $Br_2$  and BrCl triggers the ozone destruction leading to ODEs given the availability of light for photolysis.

emission of  $\mathrm{Br}_2$  as described in the following paragraph. We depict the most relevant cycles for ODEs grouped by families in a reaction network (Fig. 1). The  $\mathrm{NO}_x$  cycle can be net-zero regarding ozone destruction and production.

In the following, we briefly describe the most relevant MESSy submodels that handle trace gas emissions and sinks. For a complete list of submodels used in this study, please consult the file switch.nml in Supplement A (EMAC\_namelists.zip). The boolean values therein indicate whether the submodels were used.

Trace gas emissions that depend on the state of the Earth system components (atmosphere, land, and ocean) are computed during the model integration in the MESSy submodel ONEMIS Kerkweg et al. (2006b). Therein, the subroutine AirSnow (Falk and Sinnhuber, 2018) handles the emission of bromine from sea ice and snow in the polar regions following the scheme suggested by Toyota et al. (2011). The dry deposition flux of  $O_3$  on first-year sea ice (FY) triggers  $Br_2$  emissions. Assuming that a considerable amount of bromide ( $Br^-$ ) has already been pushed out of multi-year sea ice (MY), recycling is limited by the amount of inorganic bromine (HBr, BrNO<sub>3</sub>, and HOBr) deposition from gas-phase. A similar assumption is used for snow on land (LS). The distribution of MY sea ice is prescribed. This scheme introduces a critical temperature  $T_{\rm crit}$  effectively acting as a seasonal limit and a critical solar zenith angle ( $\theta_{\rm crit}$ ) as a trigger that separates dark from sun-lit conditions with two molar yields ( $\phi_1$ ,  $\phi_2$ ) parameterizing in-snow photochemistry. The sea salt aerosol mass flux is computed with the LSCE scheme (Guelle et al., 2001) that applies lookup tables depending on wind speed. A detailed comparison of all available sea salt aerosol emission schemes is found in Kerkweg (2005). The corresponding bromine flux is derived by scaling this flux with a Cl to Br molar ratio of 667 and assuming half of this flux is released into the gas-phase (Yang et al., 2008).

Ocean-to-atmosphere fluxes of organic substances dissolved in seawater (e.g. dimethylsulfate (DMS), bromoform  $CHBr_3$  and dibromomethane  $CH_2Br_2$ ) (Pozzer et al., 2006) are computed by the AIRSEA submodel. The flux for each species is calculated from concentration gradients in the uppermost ocean layer and the lowermost atmosphere layer. The species-specific Henry's law constant defines the respective solubility. The corresponding concentrations of brominated very short-lived substances (VSLS) in ocean waters are taken from Wang et al. (2019b).

The submodels OFFEMIS and TNUDGE handled emissions from preprocessed 4D forcing fields (Kerkweg et al., 2006b). Emission inventories of greenhouse gases (GHGs), Ozone Depleting Substances (ODSs), and ozone precursors are based on the German Aerospace Centre (Deutsches Zentrum für Luft- und Raumfahrt, DLR) CCMI2 inventory (available until 2019). GHGs included are N<sub>2</sub>O, CH<sub>4</sub>, and CO<sub>2</sub>. ODSs include a wide range of halons, chlorofluorocarbons (CFCs), and hydrochlorofluorocarbons (HCFCs). Also included are potential ozone precursors like Volatile Organic Compounds (VOCs) and Non-Methane-Hydro-Carbons (NMHCs) from biomass and agricultural waste burning as well as in fossil fuel emissions from traffic and non-traffic consisting of VOCs (only acetylene (C<sub>2</sub>H<sub>2</sub>)), NMHCs, NO, CO, SO<sub>2</sub>, NH<sub>3</sub> plus aerosol SO<sub>4</sub> derived from SO<sub>2</sub>, organic carbon (OC) and black carbon (BC) emissions. These CCMI2 inventories are not available beyond 2019 for our model, so we substituted the missing data with 2019 emissions.

Emissions of the minor VSLS (CHCl $_2$ Br, CHCl $_2$ Br, CH $_2$ Cl $_3$ Br are based on Warwick et al. (2006). The submodel TNUDGE nudges the surface concentration climatologies of the long-lived halons CF $_3$ Br and CF $_2$ Cl $_3$ Br towards observed mixing ratios.

Dry deposition is the most important way of removing non-soluble chemical substances from the atmosphere. The corresponding dry deposition velocities ( $v^{\rm dd}$ ) are computed by the submodel DDEP (Kerkweg et al., 2006a) following the Wesely-scheme (Wesely, 1989):

$$v^{\rm dd} = \frac{1}{r_a + r_b + r_c},\tag{1}$$

with aerodynamical resistance  $(r_a)$  driven by atmospheric turbulence, quasi-laminar resistance  $(r_b)$  driven by molecular diffusion and turbulence, and surface resistance  $(r_c)$ . For most species, there are no observations of  $r_c$ . Estimates are calculated based on the Henry's law coefficient  $(H_i)$ , a reactivity rate coefficient  $(R_{dry}^i)$ , and  $r_c$  of  $O_3$  and sulfur dioxide  $(SO_2)$ :

$$r_{c,\text{snow}}^{i} = \frac{1}{\frac{H_{i}}{r_{c,\text{snow}}^{\text{SO}_{2}} \cdot 10^{5}} + \frac{R_{\text{dry}}^{i}}{r_{c,\text{snow}}^{\text{O}_{3}}}}$$
(2)

with  $r_{c,\,{
m snow}}^{{
m O}_3}=2000\,{
m s}\,{
m m}^{-1}$  and  $r_{c,\,{
m snow}}^{{
m SO}_2}=1\,{
m s}\,{
m m}^{-1}.$ 

Wet deposition of soluble species is calculated via the SCAV submodel (Tost et al., 2006). It considers convective and large-scale precipitation separately as well as three modes of scavenging from ice-, liquid- and gas-phase.

#### 155 2.2 Observational data

For evaluating our model results, we use surface ozone VMR from a network of Arctic monitoring sites, satellite observations of BrO column VCD from the TROPOspheric Monitoring Instrument (TROPOMI) onboard the European Space Agency (ESA)

**Figure 2.** Locations of the Arctic ozone monitoring sites used in this work. The Research vessel Polarstern's drift track during the MOSAiC expedition (March-April-May 2020) is indicated in blue-red-orange, respectively. A spring 2020 mean sea ice age distribution from the Integrated Climate Data Center (ICDC) age of sea ice product (Tschudi et al., 2024) is shown.

Copernicus Sentinel-5 Precursor satellite, and atmosphere and trace gas observational records from the Multidisciplinary drifting Observatory for the Study of Arctic Climate (MOSAiC) expedition (2019/20). The ozone monitoring sites and MOSAiC drift track locations' are shown in Fig. 2.

#### 2.2.1 Ozone monitoring sites

There are several monitoring sites providing long- and short-term  $\chi_{\rm O_3}$  records in the Arctic as listed in Table 1. For studying the similarity between modeled and observed  $\chi_{\rm O_3}$ , and possible changes due to a changing climate, we use long-term data from the National Oceanic and Atmospheric Administration (NOAA) atmospheric monitoring station Barrow at Utqiaġvik, Alaska (USA) (McClure-Begley et al., 2024) spanning 5 decades, the over 3 decade long ozone record from Mt. Zeppelin, Ny-Ålesund (Spitsbergen), and almost 30 years of observations from Alert, Nunavut (Canada). We supplement these with data records from sites located at Eureka, Nunavut (Canada), and Nord and Summit both located in Greenland (Denmark). All data, except for Barrow station, have been downloaded from the EBAS website (Norsk institutt for luftforskning (NILU)) in 1-hourly resolution. All model output was interpolated to the sites' locations using the MESSy submodel SCOUT. No vertical interpolation to the stations' exact altitudes was performed.

# 2.2.2 TROPOMI

170

The ESA Copernicus Sentinel 5-Precursor satellite was launched in October 2017 with a designed lifetime of 7 years. It is on a sun-synchronous orbit with local time of ascending node is at 13:30 h. The TROPOMI instrument has a near-nadir resolution

**Table 1.** Ozone monitoring sites in the Arctic and data periods used in this work.

| Station                   | Period    | Coordinates        |          |          |  |
|---------------------------|-----------|--------------------|----------|----------|--|
|                           |           | latitude longitude |          | altitude |  |
|                           |           | (° N)              | (° E)    | (m)      |  |
| Alert <sup>†</sup>        | 1988–2013 | 82.499             | -62.341  | 204.0    |  |
| Eureka <sup>†,‡</sup>     | 2018-2024 | 79.983             | -85.95   | 610.0    |  |
| Nord †                    | 2001-2022 | 81.600             | -16.67   | 20.0     |  |
| Utqiaġvik <sup>‡</sup>    | 1973-2024 | 71.323             | -156.611 | 11.0     |  |
| Summit <sup>†</sup>       | 2000-2024 | 72.578             | -38.459  | 3238.0   |  |
| Mt. Zeppelin <sup>†</sup> | 1989–2024 | 78.910             | 11.888   | 474.0    |  |

Data source: † NILU - EBAS, † NOAA - Global Monitoring Laboratory. More details under *data availability*.

of  $3.5 \times 7 \,\mathrm{km^2}$  ( $3.5 \times 5.5 \,\mathrm{km^2}$  since July 2022) and a swath width of  $2,600 \,\mathrm{km}$ . Its level 2 data products include  $O_3$  (total and tropospheric column, profile),  $NO_2$  (total and tropospheric column),  $SO_2$  (total column), and carbon monoxide CO (total column). For this study, PO total and tropospheric columns have been retrieved with an optimized and adapted Differential Optical Absorption Spectroscopy (DOAS) retrieval algorithm that was developed for earlier satellite missions (Seo et al., 2019, 2020). To isolate the tropospheric from the total column, a stratospheric PO climatology (Theys et al., 2009) has been used. As no total column air-mass factor (AMF) for TROPOMI is currently available, a stratospheric AMF has been applied for the total columns. For the tropospheric columns, a simplified approach was used assuming a bright surface (albedo of 0.9) and a PO surface layer of PO with thickness. This affects the retrieval over dark surfaces, e.g. ocean, and boreal forest, by reducing the amount of reconstructed PO VCD (Choi et al., 2012).

In Fig. 3 TROPOMI monthly mean BrO total and tropospheric column VCD are shown for April 2019. The tropospheric column VCD (Fig. 3(b)) indicates BrO enhancement over the whole Arctic Ocean but most prominently over the Canadian archipelago and the Kara/Laptev Sea (BrO VCD =  $(3-4) \cdot 10^{13}$  molec cm<sup>-2</sup>).

# 2.2.3 MOSAiC expedition

The MOSAiC expedition was the largest scientific expedition in the Arctic to date. From September 2019 to October 2020, the German research vessel Polarstern drifted with the sea ice in the Central Arctic. In the course of this mission, a multitude of interdisciplinary experiments were conducted, including the measurement of meteorological conditions (Jozef et al., 2023) as well as O<sub>3</sub> (Angot et al., 2022), Br and BrO(Mahajan, 2022). The position of Polarstern during Spring 2020 is shown in Fig. 2.

The modeled surface temperature and BL height interpolated to the drift track are in general in good agreement with the observations (Fig. 4). This demonstrates that nudging with ERA5 reproduces the observed weather and meteorological conditions. Below  $-10\,^{\circ}$ C, however, modeled surface temperatures becomes apparently warmer than observed (2K 

**Figure 3.** TROPOMI BrO VCD for April 2019. Stratospheric BrO from a stratospheric climatology by Theys et al. (2009) was subtracted from TROPOMI total column to derive the tropospheric contribution. (a) Total column; (b) Tropospheric column.

(Fig. 4(c)). This warm bias increases with decreasing temperatures and is most prominent when BL heights are below 100 m. This could be caused by the ERA5 nudging data in combination with the relatively low vertical model resolution in the BL. However, nudging only starts above model level 8. Wang et al. (2019a) have identified a regionally varying bias in both ERA-interim and ERA5 that increases at low temperatures (most notably below  $-20^{\circ}$ C) compared with buoy observations which is similar to the bias observed in our model experiments.

# 3 Model parameter sensitivity and setup

In this section, we describe the sensitivity of the AirSnow algorithm to critical parameters and boundary conditions. We introduce a more realistic multi-year sea ice concentration derived from the Integrated Climate Data Center (ICDC) age of sea ice product (Sect. 3.1) and implement a sigmoidal relaxation of the temperature threshold (Sect. 3.2). We test the performance of the AirSnow algorithm and find the best model setup (Sect. 3.3). To make sure that the model is not over-tuned, we exclusively use the period January–July 2019 for improving the model skill in terms of  $O_3$  at observation sites.

All model experiments are listed in Table 2. In contrast to the release version in MESSy v2.55.2, we include two critical bug fixes concerning the temperature (conversion from Celsius to Kelvin) and solar zenith angle (sign flip) thresholds. We have already reported and fixed these in the release candidate of MESSy v2.56.

By default, bromine prescribed from the sea salt aerosol mass flux was not treated as chemical tracer, referred to as *diagnostic* in Table 2. We refer to the inclusion of Br<sub>2</sub> to the tendency of the respective chemical tracer at the lowest atmospheric level as *interactive*.

Figure 4. Comparison of meteorological observations during the MOSAiC expedition with EMAC model results from the reference simulation mapped to the MOSAiC drift track. (a) Surface temperature; (b) BL height; (c) Temperature difference ( $\Delta T = T_{\rm EMAC} - T_{\rm MOSAiC}$ ), observed BL height indicated by color. The dashed red line represents  $T_{\rm crit} = -15\,^{\circ}{\rm C}$ .

Frozen freshwater lakes were originally included in the model's SIC, which caused false BEs around the Great Lakes as these were treated like first-year sea ice. We therefore, first excluded these areas by masking them out in the SIC and later treated them like land snow.

#### 215 3.1 Multi-year sea ice cover fraction

By construction, the amount of  $\mathrm{Br}_2$  emitted using the Toyota scheme is sensitive to the assumed age of sea ice. Therefore, we expect more  $\mathrm{Br}_2$  emissions from regions with a large first-year sea ice concentration (FYSIC). Falk and Sinnhuber (2018) derived a multi-year sea ice concentration (MYSIC) from ERA-interim with the assumption of a static multi-year sea ice distribution computed from the SIC at the seasonal minimum of the previous year. We used the same assumption to derive MYSIC from ERA5 SIC and compared this with an age of sea ice (AoSI) product provided by the Integrated Climate Data Center (ICDC) (Tschudi et al., 2024).

ICDC AoSI uses sea ice drift data from satellite observations to assign an age to the individual ice floats for which drift trajectories are computed. Each grid cell with at least 15% SIC is treated as a Lagrangian particle and traced every week. The co-existence of ice of varying ages in one grid cell prefers the survival of the older ice because younger and thinner sea ice deforms and melts more easily. This causes an overestimation of the multi-year sea ice cover.

**Table 2.** List of model experiments. We use MESSy v2.55.2 including two critical bug fixes in the AirSnow mechanism that have been integrated into the MESSy release candidate. The MECCA chemistry mechanism CCMI2-base-01 has been applied for all experiments. MYSIC based on ICDC Age of Arctic Sea Ice has been made available to the MESSy community. Sea salt bromine was not treated as chemical tracer by default, here referred to as *diagnostic*. Frozen lakes are included in the model's SIC and were either masked out or treated like land snow. Experiment sfa002 has been submitted to the Arctic Bromine Model Intercomparison project.

| Exp.   | Period          | AirSnow | MYSIC | $T_{ m crit}$            | Sea salt bromine | notes                     |
|--------|-----------------|---------|-------|--------------------------|------------------|---------------------------|
| ref    | 2019-01-2020-07 | off     | -     | -                        | diagnostic       |                           |
| sfa002 | 2019-01-2020-07 | on      | ERA5  | $-15^{\circ}\mathrm{C}$  | diagnostic       | model intercomp.          |
| sfa008 | 2019-01-2019-07 | on      | ERA5  | $-15^{\circ}\mathrm{C}$  | diagnostic       | + frozen lakes masked     |
|        |                 |         |       |                          |                  | + sigmoidal $T_{ m crit}$ |
| sfa010 | 2019-01-2019-07 | on      | ICDC  | $-15^{\circ}\mathrm{C}$  | diagnostic       | + frozen lakes masked     |
|        |                 |         |       |                          |                  | + sigmoidal $T_{ m crit}$ |
| sfa011 | 2019-01-2019-07 | on      | ICDC  | $-10^{\circ}\mathrm{C}$  | diagnostic       | + frozen lakes masked     |
|        |                 |         |       |                          |                  | + sigmoidal $T_{ m crit}$ |
| sfa012 | 2019-01-2019-07 | on      | ICDC  | $-2.5^{\circ}\mathrm{C}$ | diagnostic       | + frozen lakes masked     |
|        |                 |         |       |                          |                  | + sigmoidal $T_{ m crit}$ |
| sfa013 | 2019-01-2019-07 | on      | ICDC  | $-10^{\circ}\mathrm{C}$  | diagnostic       | + frozen lakes like LS    |
|        |                 |         |       |                          |                  | + sigmoidal $T_{ m crit}$ |
| sfa017 | 2019-01-2020-07 | on      | ICDC  | $-10^{\circ}\mathrm{C}$  | interactive      | + frozen lakes like LS    |
|        |                 |         |       |                          |                  | + sigmoidal $T_{ m crit}$ |

To derive a MYSIC, we summed over all ICDC age classes  $\geq 1\,\mathrm{year}$  and average the bi-weekly data over one month. The original resolution of  $12.5 \times 12.5\,\mathrm{km}$  was then remapped onto the target resolution (e.g. T42, T106, Appendix Fig A1).

Compared to ERA5-derived MYSIC, ICDC AoSI indicates distinctively less multi-year sea ice in the Central Arctic Ocean and the Canadian archipelago but more east of Greenland, north of Spitsbergen and Alaska (Fig. 5(a)). To identify which of the two MYSIC estimate is more realistic, we computed the total area covered by multi-year sea ice. The area derived from ERA5 is consistently  $\sim 30\%$  larger than from ICDC AoSI over the period 1980 to 2020 (Appendix Fig. A2(a)). The total area of multi-year sea ice based on ICDC AoSI amounts on average to  $2.5 \cdot 10^6 \, \mathrm{km}^2$  in the 2020s, which is consistent with the range  $(0.5-2.9) \cdot 10^6 \, \mathrm{km}^2$  given by Regan et al. (2023) for the time period 2009–2019.

We conducted two model experiments that differ only in the applied MYSIC (ERA5, ICDC AoSI) and found that the lower the MYSIC in a grid cell the more  $\mathrm{Br}_2$  and the less  $\mathrm{BrCl}$  emissions from ice and snow are simulated (Fig. 5(b)). The opposite is true for grid cells with larger MYSIC. This means that strong local sources of  $\mathrm{Br}_2$  due to small-scale sea ice inhomogeneities may appear smeared out in our applied model resolution.

For our model experiments, we computed BrO VCD from  $\chi_{\rm BrO}$  and used a dynamical tropopause metric based on the model potential vorticity (PV) at  $3.5\,\rm PVU$  ( $1\,\rm PVU=10^{-6}\,m^2\,s^{-1}\,K\,kg^{-1}$ ) to separate tropospheric and stratospheric columns. We

Figure 5. Influence of MYSIC on AirSnow emission fluxes. (a) MYSIC difference ICDC AoSI-ERA5. (b)  $\mathrm{Br}_2$  and (c)  $\mathrm{BrCl}$  flux difference between a model experiments with ICDC AoSI (sfa010) and ERA5 (sfa008).

sampled the modeled total and tropospheric VCD at  $13-14\mathrm{h}$  local time and calculated monthly averages to compare with the satellite retrieval. The mean modeled stratospheric contribution to the total BrO VCD in April was rather uniform in all experiments and amounts to  $(2.5-3)\cdot 10^{13}\,\mathrm{molec\,cm^{-2}}$  (Appendix Fig. B3).

The TROPOMI BrO VCD for April 2019 (Fig. 3(a, b)) indicates hotspot regions over the Canadian archipelago and the Kara/Laptev Sea. This pattern is well reproduced in experiments with ICDC AoSI-derived MYSIC (Appendix Fig. B4(c, d)). The ERA5-derived MYSIC shows an additional hotspot at the Alaskan coast (Appendix Fig. B4(a, b)) which is absent in the satellite VCD. Therefore, we conclude that MYSIC derived from ICDC AoSI is the better choice.

## 3.2 Temperature threshold

The threshold temperature  $T_{\rm crit}$  constrains the occurrences of BEs in time and space during the spring. Toyota et al. (2011) noted that there is no  $T_{\rm crit}$  that optimizes modeled ozone at all monitoring sites simultaneously. We propose that relaxing this temperature threshold might improve the agreement without implementing a detailed snow microphysics and snow chemistry scheme (Toyota et al., 2014) in a global CCM. For this purpose, we implemented a sigmoidal temperature dependency of the form

$$f_{\text{crit}}(T) = \frac{1}{1 + e^{T - T_{\text{crit}}}},\tag{3}$$

with T the surface temperature in a given grid cell. The  $\mathrm{Br}_2$  and  $\mathrm{BrCl}$  emission fluxes are then scaled with  $f_{\mathrm{crit}}(T)$  and reach 10% and 90% of the maximum flux at  $T_{\mathrm{crit}} \pm 2.2\,^{\circ}\mathrm{C}$ , respectively.  $\mathrm{Br}_2$  emissions thus already increase where temperatures are slightly above  $T_{\mathrm{crit}}$ , most notably in coastal regions.

We then looked at model results at  $T_{\rm crit} \in \{-15, -10, -2.5\}$  °C.  $T_{\rm crit} = -2.5$  °C means allowing bromine emission also at temperatures around the freezing point of freshwater. With a higher threshold temperature, the modeled  $Br_2$  fluxes consistently show an increase over first-year sea ice (Fig. 6(d, g)). Emissions from multi-year sea ice regions, indicated by the BrCl fluxes (Fig. 6(c, h)), also display an increase but remain two orders of magnitude lower than emissions from first-year sea ice

Figure 6. Comparison of modeled bromine emissions due to BEs and tropospheric BrO VCD. Flux integral of (a, d, g) Br<sub>2</sub> and (b, e, h) BrCl and monthly mean tropospheric BrO VCD at  $13-14\,\mathrm{h}$  local time (c, f, i) for April 2019. The EMAC total column BrO has been split into tropospheric and stratospheric contributions using the modeled PV-tropopause height on hourly basis. (a–c)  $T_{\rm crit}=-15\,^{\circ}\,\mathrm{C}$  (sfa010); (d–f)  $T_{\rm crit}=-10\,^{\circ}\,\mathrm{C}$  (sfa011); (g–i)  $T_{\rm crit}=-2.5\,^{\circ}\,\mathrm{C}$  (sfa012).

regardless of the choice of  $T_{\rm crit}$ . At  $T_{\rm crit} = -2.5\,^{\circ}{\rm C}$ , the model predicts  ${\rm Br}_2$  emissions in the Gulf of Bothnia, the White Sea, and the Sea of Okhotsk (Fig. 6(g)), though this does not result in notably enhanced  ${\rm BrO~VCD}$  (Fig.6(i)).

At  $T_{\rm crit}=-2.5\,^{\circ}{\rm C}$  the BrO VCD in the hot spot regions is closest to the TROPOMI VCD (Fig. 3(b)), though more enhanced BrO VCDs are found at the coast east of Greenland and East Siberia than observed. As indicated above, this could be a relict of the relatively low model resolution and associated land-sea mask that does not resolve all topographic features and congruent sub-grid temperature variance. Possibly, the stratospheric BrO climatology and simple AMF used in the TROPOMI retrieval also plays a role for the absolute values. Overall, our modeled BrO VCD in 2019 agrees best with satellite observations (Fig. 3(b)) in both amount and spatial pattern of tropospheric BrO VCD at  $T_{\rm crit}=-2.5\,^{\circ}{\rm C}$ .

#### 3.3 Model skill and best setup

To decide on the best model setup, we evaluated the model skill in terms of the coefficient of determination of the linear regression (squared Pearson correlation coefficient,  $R^2$ ) and root-mean-square-error (RMSE) at the Arctic ozone monitoring sites (Sect. 2.2.1) for spring 2019.

The resulting  $R^2$  and RMSE for 2019 at Utqiagvik, Eureka, Summit, and Zeppelin are listed in Table A1. The closer  $R^2$  is to 1, the better the correlation. The closer the RMSE is to 0, the smaller the difference between the observed and modeled time series. For the corresponding histograms with fitted linear regression curves, see Appendix Fig. A4.

We confirmed that there is not a single combination of parameters that optimizes both  $R^2$  and RMSE for all sites. The model experiment with MYSIC from ICDC AoSI and  $T_{\rm crit}=-15\,^{\circ}{\rm C}$  (sfa010) performs best with respect to  $R^2$  ( $\langle R^2 \rangle = 0.198$ ). The experiments with  $T_{\rm crit}=-10\,^{\circ}{\rm C}$  (sfa011, sfa017) are best in terms of RSME ( $\langle {\rm RMSE} \rangle = 0.365$ ). The treatment of freshwater lakes like land snow decreases the model skill slightly compared to the experiments where these are excluded (compare (sfa013, sfa017) with (sfa010, sfa011)). We will need to investigate this further. An experiment (sfa012) with  $T_{\rm crit}=-2.5\,^{\circ}{\rm C}$  performs well at Zeppelin, but displayed lower modeled  $\chi_{\rm O_3}$  than observed at Eureka and Villum research station Nord in May and June (Appendix Fig. B2). At higher spatial resolution ( $15\times15\,{\rm km}$ ), Gong et al. (2025) have shown that, when using an ODE terminator defined by snowmelt and additional bromine emission from multi-year sea ice, observed and modeled  $T_{\rm crit}=-2.5\,^{\circ}{\rm C}$ , but was underestimated at Villum research station Nord. This suggests that BEs near Zeppelin may persist at relatively high temperatures as compared to other ice-covered regions for unidentified reasons.

Based on these quantitative results, we decided on the setup with MYSIC derived from ICDC AoSI,  $T_{\rm crit} = -10\,^{\circ}{\rm C}$ , and  ${\rm Br_2}$  emissions from sea salt enabled (sfa017). As stated above, tropospheric BrO VCD showed a qualitatively better agreement with  $T_{\rm crit} = -2.5\,^{\circ}{\rm C}$ .

#### 4 Results

275

280

Our special focus lies on the spring 2020 for which ozone monitoring station records, TROPOMI satellite observations, and data from the MOSAiC expedition are available. In this section, we show the surface ozone VMR time series at different sites for our best model setup and compute probability density functions (PDFs) of χ<sub>O3</sub> from observation and model results to look for indications of a possible climate change impact on ODEs (Sect. 4.1). We use the MOSAiC data to judge the overall model skill in the Central Arctic qualitatively (Sect. 4.2) and compare the modeled pan-Arctic pattern of ozone depletion and BrO VCD with TROPOMI retrieved BrO VCD (Sect. 4.3).

#### 4.1 Arctic ozone monitoring sites

#### **4.1.1** Ozone time series 2019/20

For the Arctic ozone monitoring sites Eureka, Nord, Summit, Utqiagʻvik, and Zeppelin, we show the 2019/20 composite time series of modeled (ref, sfa017) and observed  $\chi_{O_3}$  in Fig. 7. We find that the effect of AirSnow on ozone VMR is confined

temporally. As soon as the trigger conditions for ODEs are not satisfied anymore, O<sub>3</sub> VMR returns quickly to the values of the reference simulation. It is evident that AirSnow qualitatively improves the EMAC model capabilities of capturing tropospheric ozone in the Arctic spring at all sites. Key features like the late April to early May ODE in 2019 at Eureka and Utqiaġvik and the late March to late April ODEs at Utqiaġvik in 2019 match reasonably well in their timing but not in the observed strength of the ozone depletion. An ODE in March 2019 at Eureka station coinciding with a pronounced dip in ozone at Summit is not reproduced. The EMAC model generally underestimates χ<sub>O3</sub> in Arctic winter. Following Helmig et al. (2007), Falk and Sinnhuber (2018) have shown that a higher surface resistance of ozone over ice and snow (r<sub>c</sub> = 10000 sm<sup>-1</sup>) could improve this. In 2020, especially in March and May, ODEs at Eureka and Utqiaġvik were captured less well, also reflected by a reduced model skill compared to 2019 (see Tab. B1). This was potentially caused by the recycling of 2019 emission inventories. As a secondary pollutant, tropospheric ozone depends on precursor substances like NO<sub>x</sub>, CO, and VOCs. COVID-19 policies reduced their emissions which had a measurable effect on tropospheric ozone (Weber et al., 2020; Venter et al., 2020; Steinbrecht et al., 2021).

Beyond Arctic springtime, we found peak ozone VMR at Utqiagʻvik in June 2019 and 2020 that are much larger than observed. These peak values are likely caused by large wildfires raging within the Arctic circle in 2019 (Descals et al., 2022). These usually contribute to ozone precursors and cause episodes of enhanced tropospheric ozone (Cofer et al., 1990; Lindskog et al., 2007; Karlsson et al., 2013). However, Baker et al. (2016) have reported that attenuation of solar radiation and aging of aerosols are potentially underestimated in chemistry transport models causing photolyis rates and therefore in situ ozone production to be overestimated compared to observations. For reference, surface ozone time series including only the bug fixes are shown in Appendix Fig. B1.

# 4.1.2 Ozone climatology

315

330

We assume that the amount of Br<sub>2</sub> emitted and, subsequently, the strength of ODEs should be highest for a high FYSIC (Falk and Sinnhuber, 2018). Around 2007, the amount of multi-year sea ice in the Arctic prominently dropped (Regan et al., 2023) (Appendix Fig. A2). Hence, we would expect an increased occurrence of ODEs after 2007. To test this hypothesis, we divided the observational datasets with long-term records into two periods: (1) start of record until the end of 2007 and (2) 2008 until the end of each record. For these two periods, we computed PDFs of observed χ<sub>O3</sub> as normalized histograms with 1 ppbv
binning (Fig. 8). The Arctic ozone monitoring sites with long-term records are Alert, Utqiaġvik, and Zeppelin. Error bars denote the year-by-year variance of the observational data using standard deviation.

In March, observations at Alert and Zeppelin display close-to-normal distributions peaking between  $40-45\,\mathrm{ppbv}$  with a small tail towards low  $\chi_{\mathrm{O_3}}$ . These tails become larger throughout April and May and transform the distribution into a close-to-equal distribution which can be interpreted as an ODE fingerprint. The distribution at Utqiaġvik shifts to a close-to-equal distribution already in March owing to its more southern location. The close-to-equal distribution of  $\mathrm{O_3}$  VMR during the ODE season is in line with previously identified temporally varying frequency distributions of  $\chi_{\mathrm{O_3}}$  in ship (Jacobi et al., 2010) and airplane (Ridley et al., 2003) expeditions in the Central Arctic.

Figure 7. Time series comparing observational records of  $\chi_{O_3}$  at Arctic ozone monitoring sites with EMAC model output (ref, sfa017) interpolated to the station location. The periods March–May are highlighted in light yellow. The dashed red line indicates the 5 ppbv threshold for ODEs. (a) Eureka; (b) Nord; (c) Summit; (d) Utqiagʻvik; (e) Zeppelin.

In the observational period (2), all sites' distributions display a significant increase of higher  $\chi_{O_3}$  compared to period (1) in spring. In period (2) data at Utqiagvik show the onset of a return to a close-to-normal distribution already in May compared to period (1). This means that conditions became less favorable for ODEs in recent decades. In April and May, the ODE

**Figure 8.** Histogram of  $\chi_{O_3}$  for EMAC experiments sfa017 and ref (2019/20) compared to long-term observations separated into two periods: start of record to the end of 2007 and 2008 to end of record. (a) Alert; (b) Utqiagvik; (c) Zeppelin.

bins  $(\chi_{O_3} \le 10 \,\mathrm{ppbv})$  at all stations are significantly (more than  $1\sigma$ ) less populated in time period (2). Especially, at Zeppelin almost no ODEs occur in period (2) while atmospheric background  $O_3$  concentrations apparently have increased. Although the instrumental uncertainty in earlier years is usually larger, these data imply a significantly less frequent occurrence of strong or continuous ODEs in recent decades. This means that either the role of bromine emissions from FYSIC in promoting ODEs is overstated, or that other climate-sensitive factors are counteracting the observed increase in BrO VCD (Bougoudis et al., 2020) in terms of ozone depletion.

The modeled distributions differ from observations beyond the expected year-to-year variance. The reference simulation follows a normal distribution for almost all sites and months. Only at Utqiagvik in March a slight tendency towards an equal distribution can be found. While generally too high,  $\chi_{\rm O_3}$  at Zeppelin is too low in March. Using AirSnow, the distributions get closer to observations for some months and sites. The best agreement is achieved at Alert and Zeppelin in April and Utqiagvik in March. However, a general shift towards lower ozone at all locations leads to an underestimation of the frequency of  $\chi_{\rm O_3}$  becoming greater than 37 ppbv. In March, there is only little change at Alert, Utqiagvik, and Zeppelin. In May, the PDF for Alert shifts towards lower values and is more equally distributed. We conclude that  $\chi_{\rm O_3}$  distributions are captured more correctly, when accounting for the bromine emissions from ice and snow.

**Figure 9.** Evaluation of EMAC experiments (ref, sfa002, sfa017) interpolated onto MOSAiC drift track. Compared to MOSAiC observations. (a) O<sub>3</sub>; (b) BrO. Note that  $\chi_{\rm BrO}$  were not present in the reference experiment.

#### 4.2 Ozone and bromine monoxide in the Central Arctic

In Sect. 2.2.3, we found that modeled surface temperature and BL agree reasonably well with observations during the MOSAiC expedition. In the following, we will compare modeled  $\chi_{O_3}$  and  $\chi_{BrO}$  with in situ observations during the MOSAiC expedition for the leg March–May 2020.

We interpolated  $\chi_{\rm O_3}$  and  $\chi_{\rm BrO}$  for the model experiments ref, sfa002, and sfa017 onto the MOSAiC drift track and compared these with observations (Fig. 9). We found an ozone depletion of up to 20 ppbv with our best setup (sfa017), though the observed ODEs ( $\chi_{\rm O_3} \leq 5\,{\rm ppbv}$ ) in the Central Arctic are not reproduced in any of our model experiments. During the modeled partial ODEs in April, much more surface BrO is produced than observed, while tropospheric BrO VCD remains lower than observed (compare Figs. 10(b,d)). In May,  $\chi_{\rm BrO}$  is better reproduced, while  $\chi_{\rm O_3}$  remains too high.

The photochemical steady state between Br and BrO requires the presence of  $O_3$  to maintain BrO production (Fig. 1) (Hausmann and Platt, 1994; Zhao et al., 2016). As the ozone depletion on pan-Arctic scale remains incomplete in our model experiments, the enhanced  $\chi_{\rm BrO}$  at the surface is probably a direct consequence.

#### 4.3 Pan-Arctic implications

The pattern of observed and modeled BEs show a considerable year-by-year variability. In 2020, TROPOMI tropospheric BrO VCD displays an extended area of increased BrO from the northern tip of the Canadian archipelago to the Laptev Sea (BrO VCD  $\geq 4 \cdot 10^{13}$  molec cm<sup>-2</sup>). In the model experiment, this region decomposes into three hotspots: the northern tip of Greenland and the Canadian archipelago (BrO VCD =  $(2.5-4) \cdot 10^{13}$  molec cm<sup>-2</sup>), north of the Beaufort Sea, and the Laptev/Kara Sea, north of Novaya Zemliya, (BrO VCD =  $(0.5-1.5) \cdot 10^{13}$  molec cm<sup>-2</sup>) respectively. A modeled corridor of lower BrO enhancement coincides with the maximum of MYSIC. This implies a need to reconsider the assumptions regarding

**Figure 10.** Monthly mean BrO VCD at 13 – 14h local time in April 2020. The EMAC total column BrO has been split into tropospheric and stratospheric contributions using the modeled tropopause height on hourly basis. A stratospheric climatology (Theys et al., 2009) was subtracted from TROPOMI total column. (a) TROPOMI (total column); (b) TROPOMI (tropospheric column); (c) sfa017 (total column); (d) sfa017 (tropospheric column).

bromine recycling on multi-year sea ice. These findings support the conclusions of Peterson et al. (2019), indicating that regions covered by multi-year sea ice serve as a more significant source of reactive bromine than previously recognized. The modeled  $BrO\ VCDs$  are up to a factor of four lower than observed. As shown in Sect. 3, these scale with the amount of  $Br_2$  emitted.

The monthly maximum of the difference in surface ozone (Fig. 11(a)) displays major hotspots ( $\Delta O_{3_{max}} = (30-40)\,\mathrm{ppbv}$ ) in the Canadian archipelago/Baffin Bay/north of Greenland and the Laptev Sea/Eastern Arctic Ocean. These hotspots of ozone depletion colocate partly with regions of high BrO VCD (Fig. 10(d)). Ozone depletion on a pan-Arctic scale appears relatively weak on a monthly average (Fig. 11(b)) and never exceeds 17 ppbv in April compared to an ozone background of  $(29\pm8)\,\mathrm{ppbv}$ . Hence, modeled surface ozone is only depleted by  $60\pm3\,\%$  on average. This strongly suggests that ozone depletion is too weak

**Figure 11.** Monthly (a) maximum and (b) mean ozone depletion in April 2020 computed from difference between experiment sfa017 and the reference simulation.

not only along the MOSAiC drift track (Sect. 4.2) but also on larger scales. The hotspots of maximum depletion are located in the Canadian archipelago and the Central Arctic (North of the Aleutian Islands). In March and May, the major hotspot region is again located in the Canadian archipelago, while the secondary hotspots vary both in strength and location (Appendix Fig. B5).

In the mid-latitudes, an average ozone reduction of  $0-5\,\mathrm{ppbv}$  compared to the reference simulation is predicted. This average ozone reduction could be due to an advection of ozone depleted as well as bromine enriched air masses from the Arctic. In this regard, we found a pronounced reduction maximum of ozone of up to  $25\,\mathrm{ppbv}$  in the North Atlantic between Iceland and the Faroe Islands. This signal is colocated with prescribed shipping emissions of CO, NOx, NH<sub>3</sub>, and SO<sub>2</sub> in that area and indicates that chemical ozone depletion could be locally enhanced by advected bromine from the Arctic (Fernandez et al., 2024).

#### 5 Discussion

As a secondary pollutant, tropospheric ozone depends on precursor substances like  $NO_x$ , CO, and VOCs. COVID-19 policies reduced emissions of these which had a measurable effect on tropospheric ozone background (Weber et al., 2020; Venter et al., 2020; Steinbrecht et al., 2021). This likely affected our results for 2020, because the photochemical steady state between Br and BrO requires the presence of  $O_3$  to maintain BrO production (Hausmann and Platt, 1994; Zhao et al., 2016). The apparently weak colocation of  $O_3$  depletion and enhanced BrO VCD would then be a consequence of different phases of non-linear temporal evolutions in the  $O_3$  and BrO concentrations in air masses being transported under the influence of the chemistry that leads to BEs and ODEs (e.g. Hausmann and Platt, 1994). This also fits well to airborne observations that reported no or little ozone depletion in regions with enhanced BrO VCD (Ridley et al., 2003; Salawitch et al., 2010).

Toyota et al. (2014) have demonstrated in a modeling study that snow photochemistry in the photic zone, which is leading to the formation of HOBr, contributes more to  $Br_2$  emissions than reactions in the snow skin layer. Following these results, Zhai

et al. (2023) suggested parameterizing emissions from deeper snow layers in dependency of the solar zenith angle and showed that these emissions are important to interpret bromide records in Greenlandic ice cores. Such emissions are currently neglected in our model. Beside the parameterization of snow photochemistry, Zhai et al. (2023) also introduced a more physically oriented criterion relating snow albedo and snowmelt adapted by Gong et al. (2025) allowing for a  $T_{\rm crit}=-2.5\,^{\circ}{\rm C}$ . Snowmelt probably is the final terminator of BEs (Burd et al., 2017), but experiments have shown that snow-metamorphism under non-melting temperature gradients already reduces the emission potential of bromine from snow (Edebeli et al., 2020). Peterson et al. (2019) concluded, that regions covered by multi-year sea ice serve as a more significant source of reactive bromine than previously recognized. This suggests that the assumptions of an infinite bromide source and instantaneous recycling may need to be revised. Including a multi-layer snow model with snow metamorphism and explicit snow chemistry could potentially improve the prediction of the bromine emission.

For modelling the removal of trace gases from the atmosphere, dry and wet deposition processes are essential. Conceptually, the associated dry deposition resistances parameterize the uptake of trace gases on different surfaces (Wesely, 1989). Therefore, dry deposition has to be reevaluated when surface reactions are explicitly included or parameterized. For ozone, Barten et al. (2023) reported a high median dry deposition resistance ( $r_c \propto 20000\,\mathrm{sm}^{-1}$ ) over the Central Arctic sea ice which is one order of magnitude larger than what we applied in our simulations ( $r_c = 2000\,\mathrm{sm}^{-1}$ ). Following Helmig et al. (2007), Falk and Sinnhuber (2018) had shown an improved agreement of ozone VMR between model (MESSy v2.52) and observation for  $r_c = 10000\,\mathrm{sm}^{-1}$ . As shown by (Luhar et al., 2018; Pound et al., 2020), dry deposition to the ocean can be improved by considering iodine reactions and emission from the upper ocean layer. In addition, Benavent et al. (2022) have shown the importance of iodine chemistry on the loss of tropospheric ozone during the MOSAiC campaign.

Photochemistry could be underestimated in twilight conditions above surfaces with high albedo, as j-values are usually only computed for  $\theta_{\odot} \leq 87^{\circ}$ . For a relatively low surface albedo of 0.3, Lary and McQuaid (1991) had shown that multi-scattering at  $80^{\circ} \leq \theta_{\odot} \leq 100^{\circ}$  is non-negligible even for stratospheric  $NO_x$  and  $O_3$  chemistry. At Halley station (Antarctica), the observed ratio of upwelling to downwelling flux for j-values of selected molecules averaged 0.98 under cloudy conditions, regardless of wavelength or solar zenith angle. Under less cloudy conditions, this ratio increased with wavelength when the solar zenith angle exceeded 75° (Jones et al., 2008). At the same time, modeled surface albedo could be underestimated over sea ice with little snow cover. At a snow depth of  $\geq 1\,\mathrm{cm}$  and for  $T 

**Figure A1.** Maps of multi-year sea ice cover in April. (i) Estimated from ERA5 sea ice cover at seasonal low of the previous year, (ii) ICDC Age of Sea Ice product (T106). The comparison between T42 and T106 illustrates the potential benefit of a higher model resolution for resolving inhomogeneity in the characteristics of sea ice such as its age. (a) 2019; (b) 2020.

**Figure A2.** Annual (ERA5) and monthly (ICDC) mean time series of the total area covered by multi-year sea ice in the Arctic. MYSIC total areas estimated the two methods differ by 30%.

**Figure A3.** Comparison of modeled bromine emissions due to BEs. Flux integral of (a, c) Br<sub>2</sub> and (b, d) BrCl for April 2019 with MYSIC based on (a, b) ERA5 SIC (sfa008); (c, d) ICDC AoSI (sfa010).

**Table A1.** Model skill ( $R^2$  and RMSE) at different Arctic ozone monitoring sites for different model experiments for spring 2019. For the respective 2d histograms and regressions see Appendix Fig. A4.

| Experiment | Test           | Utqiaġvik | Eureka | Summit | Zeppelin |
|------------|----------------|-----------|--------|--------|----------|
| ref        | $R^2$          | 0.19      | 0.01   | 0.22   | 0.01     |
|            | RMSE           | 0.45      | 0.96   | 0.18   | 0.34     |
| sfa002     | $\mathbb{R}^2$ | 0.28      | 0.16   | 0.30   | 0.00     |
|            | RMSE           | 0.36      | 0.67   | 0.17   | 0.33     |
| sfa008     | $R^2$          | 0.29      | 0.17   | 0.29   | 0.00     |
|            | RMSE           | 0.36      | 0.66   | 0.17   | 0.33     |
| sfa010     | $R^2$          | 0.32      | 0.19   | 0.28   | 0.00     |
|            | RMSE           | 0.36      | 0.65   | 0.17   | 0.34     |
| sfa011     | $\mathbb{R}^2$ | 0.28      | 0.17   | 0.30   | 0.01     |
|            | RMSE           | 0.34      | 0.63   | 0.17   | 0.32     |
| sfa012     | $\mathbb{R}^2$ | 0.23      | 0.03   | 0.26   | 0.18     |
|            | RMSE           | 0.32      | 0.74   | 0.18   | 0.31     |
| sfa013     | $\mathbb{R}^2$ | 0.28      | 0.17   | 0.28   | 0.01     |
|            | RMSE           | 0.34      | 0.64   | 0.17   | 0.32     |
| sfa017     | $\mathbb{R}^2$ | 0.28      | 0.17   | 0.28   | 0.01     |
|            | RMSE           | 0.33      | 0.63   | 0.18   | 0.32     |

Figure A4. Sensitivity test. 2D histogram and linear fit of  $\chi_{O_3}^{\text{model}}$  vs  $\chi_{O_3}^{\text{obs}}$  for 2019, March–May. The respective root-mean-square error (RMSE) and coefficient of determination  $R^2$  are listed in Appendix Table A1. From top to bottom: model experiments as given in Tab. 2. Ozone monitoring sites from left to right: Utqiaġvik, Eureka, Summit, and Zeppelin.

# **Appendix B: Results**

**Figure B1.** Time series comparing observational records of  $\chi_{\rm O_3}$  at Arctic ozone monitoring sites with EMAC model output (ref, sfa002) interpolated to the station location. The periods March–May are highlighted in light yellow. The dashed red line indicates the 5 ppbv threshold for ODEs. (a) Eureka; (b) Nord; (c) Summit; (d) Utqiaġvik; (e) Zeppelin.

Figure B2. Time series comparing observational records of  $\chi_{\rm O_3}$  at Arctic ozone monitoring sites with EMAC model output (sfa012, sefa017) interpolated to the station location. The periods March–May are highlighted in light yellow. The dashed red line indicates the 5 ppbv threshold for ODEs. (a) Eureka; (b) Nord; (c) Summit; (d) Utqiaġvik; (e) Zeppelin. For early spring (March to April), no differences are expected. The difference in  $T_{\rm crit}$  will only affact May if surface temperatures rise above  $-10\,^{\circ}{\rm C}$ .

**Table B1.** Model skill ( $R^2$  and RMSE) at different Arctic ozone monitoring sites for different model experiments in 2020.

| Experiment | Test  | Utqiaġvik | Eureka | Nord | Summit | Zeppelin |
|------------|-------|-----------|--------|------|--------|----------|
| ref        | $R^2$ | 0.06      | 0.03   | 0.09 | 0.17   | 0.00     |
|            | RMSE  | 0.87      | 1.17   | 0.64 | 0.17   | 0.39     |
| sfa002     | $R^2$ | 0.10      | 0.05   | 0.05 | 0.20   | 0.07     |
|            | RMSE  | 0.79      | 0.87   | 0.61 | 0.17   | 0.34     |
| sfa017     | $R^2$ | 0.11      | 0.10   | 0.01 | 0.23   | 0.15     |
|            | RMSE  | 0.71      | 0.74   | 0.62 | 0.16   | 0.30     |

Figure B3. Monthly mean stratospheric BrO column at 13 - 14 h local time in April 2019. Shown are experiments (a) sfa008, (b) sfa010, (c) sfa013, and (d) sfa017.

**Figure B4.** BrO column at 13 – 14 h local time in April 2019. (a) sfa008 (total column); (b) sfa008 (tropospheric column); (c) sfa010 (total column); (d) sfa010 (tropospheric column); (e) sfa013 (total column); (f) sfa013 (tropospheric column); (g) sfa017 (total column); (dh) sfa017 (tropospheric column). The modeled BrO total columns have been split into tropospheric and stratospheric columns using the modeled tropopause height on hourly basis.

**Figure B5.** Monthly (a) maximum and (b) mean ozone depletion in March and (c) maximum and (d) mean ozone depletion May 2020 computed from difference between experiment sfa017 and the reference simulation.

Author contributions. SF wrote and edited the manuscript, performed the model simulations, analyzed the model results for the ozone monitoring sites, bromine fluxes, and vertical column densities, and implemented and tested the model improvements. LR contributed to the analysis of MOSAiC data including the remapping of the drift track as well as pan-Arctic ozone, improvements to the setup, and identification of model bugs. BZ performed the TROPOMI analysis. AR provided the TROPOMI data. BMS proposed the project 'BromoPole' and contributed with insights and ideas. All authors contributed to the discussion and writing of the manuscript.

Competing interests. At least one of the (co-)authors is a member of the editorial board of Atmospheric Chemistry and Physics.

500

Acknowledgements. This work was funded by the Deutsche Forschungsgemeinschaft (DFG, German Research Foundation) '517648310'.

This work was performed on the HoreKa supercomputer funded by the Ministry of Science, Research and the Arts Baden-Württemberg. The authors would like to thank the German Federal Ministry of Education and Research and the German federal states (http://www.nhr-verein.de/en/our-partners) for supporting this work as part of the National High-Performance Computing (NHR) joint funding program. We gratefully acknowledge the funding by the DFG – project no. 268020496 - TRR 172, within the Transregional Collaborative Research Center "ArctiC Amplification: Climate relevant Atmospheric and SurfaCe Processes, and Feedback Mechanisms (AC)3".

The authors like to thank Patrick Jöckel (DLR) for his helpful comments and insights regarding the MESSy framework and submodel setup. Special thanks go to Ole Kirner (SimLab, KIT) and Sören Johansson (IMKASF) for valuable help for the model and machine setup. Thanks to ICDC, CEN, University of Hamburg for data support, especially Stefan Kern. We thank all those who contributed to MOSAiC and made this endeavour possible (Nixdorf et al., 2021).

We thank Jennie Thomas (CNRS, Institut des Géosciences de l'Environnement, Grenoble, France) for organizing the Arctic Bromine

Model Intercomparison project, which is a part of CATCH (the Cryosphere and Atmospheric Chemistry) and to which we contributed with parts of this work.

ChatGPT has been utilized to generate the manuscript abstract and title based on the Section "discussions and conclusions" as well as word smithing the outlook summary from a list of key points.

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
