# Peer review of "Challenges in Simulating Ozone Depletion Events in the Arctic Boundary Layer: A Case Study Using ECHAM/MESSy for Spring 2019/20"

_EGUsphere, 2025_

## Author Comment (AC1)

**Author's response to Anonymous Referee #1 (2025-08-09)**

Thank you for your comments helping to improve the overall presentation of our research results. We will address the points in our revised manuscript as follows:

**Line 67: there is an "e.g." that isn't followed by anything.**
  Has been removed

**Line 104: In the supplement files I downloaded, I did not see anything labelled "Supplement B".**
  "Supplement B" refers to the file `switch.nml` in Supplement A. This `FORTRAN` namelist file contains a list of all EMAC submodels. A boolean value (`TRUE/FALSE`) therein indicates whether they were used in our experiments. We will adapt the sentence accordingly:
  → *For a complete list of submodels used in this study, please consult the file* `switch.nml` *in Supplement A (*`EMAC_namelists.zip`*). The boolean values therein indicate whether the submodels were used.*

**Fig 1 and Section 2.1: Peroxyacetyl nitrate (PAN) is a source of NOx in the Arctic from long-range transport -- Does your model contain PAN and its decomposition to NOx?**
  The following PAN reactions are included in our chemical mechanism – see Supplement A (*mecchanism.pdf*) for the full list of reactions including MPAN (methacryloyl peroxynitrite).
    $CH_3 C(O)OO + NO_2 \rightarrow PAN$
    $PAN + OH \rightarrow HCHO + CO + NO_2 + H_2O$
    $PAN \rightarrow CH_3 C(O)OO + NO_2$
    $PAN + hv \rightarrow CH_3 C(O)OO + NO_2$

**Line 239-241 and 243-245: To support the text comparing modelled and satellite-measured BrO, can you please either add the TROPOMI VCD of BrO to Figure 6 so that we can see how it compares side by side? Or at the least, you should reference Fig 3(b) in the text here, so that the reader is pointed to where to look to compare.**
  We refer to Fig. 3(b) in the text, because Fig. 6 is already quite busy.

**Line 261: showed a better what?**
  The word *agreement* seemingly has disappeared during the final manuscript editing before submission. We add it accordingly.

**Fig 6: text font is too small in the panels.**
  We adapted the font size for better readability.

**Sec 4, first paragraph: Section 4.3 is referenced before Section 4.2. Text should be re-ordered to the flow of the paper.**
  We have reordered the sentence.

**Line 268-269: This sentence seems very similar to that at line 260-261. Does it really need repeating here?**
  We have removed the repetition in line 268-269.

**Line 280: Should "In 2000," be "In 2020," here?**
  Indeed. This is a typo and has been corrected.

Line 286: "raging within the Arctic cycle in 2019" should that be Arctic *circle*?

Indeed, another typo. It has been corrected.

Fig 7 and Fig B1: "The periods March–May are highlighted in linen" ß linen is the light yellow colour? As 'linen' isn't typically a word used for a colour (in my experience), maybe better to say 'light yellow' here.

Thank you for pointing this out. We changed the name accordingly.

Line 294-297: As you mention 2007 as the transition period, how come you chose 2000 as the cut off for the before and after time periods? Wouldn't the results after 2000 still have the pre-2007 conditions included in the average?

Thank you for pointing this out!
We had chose the cut off after 2000 to make the two subsets for all stations as equal in terms of size as possible for statistical reasons. The relatively short data set from Alert is very susceptible to a cut off around 2007±1 as only a couple of years remain after the cut off, while the longer datasets from Utqiagvik and Zeppelin are less affected. As the results in principle remain the same when choosing 2007 as the cut off year – except for an expected larger standard deviation in the second subset of Alert (Fig. 1) – we would adapt our manuscript accordingly for consistency.

[Figure]

Fig. 1: Profile histograms for Alert (a) and Utqiagvik (b), Zeppelin (c) for a cutoff year 2007.

Line 301: "The tails become larger throughout April and March" – do you mean *April and May* since you already mentioned March in the prior sentence?

       Indeed, we meant May.

Fig 9b: The MOSAIC dots look like they fall into vertical lines (e.g. on May 15 the dots span about 2-10 pptv. Is this because the uncertainty on those measurements is quite high or because it has a large diurnal cycle? What would the error bar be for each dot?

       BrO is produced in the catalytic destruction of ozone which is driven by the photolysis of $Br_2$ (or other like HOBr):

              $Br_2 + h\nu \rightarrow 2Br$

              $HOBr + h\nu \rightarrow Br + OH$

              $Br + O_3 \rightarrow BrO$

       This rapid increase in BrO following the diurnal cycle is, hence, expected.

       The BrO observational data including uncertainties is described in Benavent et al. (2022) and has been published by Mahajan (2022). The average uncertain is 0.6 pptV ranging between 0.3 and 1.0 pptV (Benavent et al. (2022), Fig. 1(c)).

Line 373: might not *be* the solution

       Corrected.

Sec 5: The Summary and Conclusion section is currently a little light on what you did in this study, and quite heavy on what others have done in the context of your future work. To shorten the latter and reduce duplication with what was already discussed in Section 4, you could simply list the future work items (snow, iodine chemistry, dry deposition). And for the former, for example, you could summarize the steps you took to improve the simulations of ODEs (e.g. sea ice and critical temperature).

       We will follow your suggestion and revise the Section accordingly in our next revision.

Fig A1: Instead of "middle", "right" and "left", can you please include labels for each panel? (e.g. "(i) for upper left, (ii) for upper middle, etc). It is somewhat difficult to determine what is what from the current caption. Please also increase font size of the panel titles.

       We improve the readability of the figure accordingly.

Fig A3: Similarly, It is not clear here that Br2 is the left and BrCl is the right, since it's not mentioned except in extremely small font on the colour scale label. Please add additional panel labels, and increase the font size in this figure.

       We increase the label size for improved readability by adding additional labels.

Fig A4: the text in this figure is unreadably small.

       The figure is meant as a supplement to Appendix Table A1 in which the respective RMSE and $R^2$ are listed. We increase the font size better readability.

**Bibliography**

Benavent, N., Mahajan, A. S., Li, Q., Cuevas, C. A., Schmale, J., Angot, H., Jokinen, T., Quéléver, L. L. J., Blechschmidt, A.-M., Zilker, B., Richter, A., Serna, J. A., Garcia-Nieto, D., Fernandez, R. P., Skov, H., Dumitrascu, A., Simões Pereira, P., Abrahamsson, K., Bucci, S., Duetsch, M., Stohl, A., Beck, I., Laurila, T., Blomquist, B., Howard, D., Archer, S. D., Bariteau, L., Helmig, D., Hueber, J., Jacobi, H.-W., Posman, K., Dada, L., Daellenbach, K. R., and Saiz-Lopez, A.: Substantial contribution of iodine to Arctic ozone destruction, Nat. Geosci., 15, 770–773, https://doi.org/10.1038/s41561-022-01018-w, 2022.

Mahajan, A.: Substantial contribution of iodine to Arctic ozone destruction - data, https://doi.org/10.17632/BN7YTZ4MFZ.1, 2022.

---

## Author Comment (AC2)

**Author's response to Anonymous Referee #2 (2025-08-24)**

Thank you for your thorough review of our manuscript and detailed comments that help to improve the overall presentation of our research results. We will address the points in our revised manuscript as follows:

**[Major comments]**

1. Since sfa011 (Tcrit = -10 deg C) shows the best overall performance from evaluation against the 2019 surface ozone data from four Arctic coastal stations, the authors have decided to use -10 deg C as Tcrit for modelling the 2020 case (sfa017). However, model run sfa012 (Tcrit = -2.5 deg C) significantly improves the 2019 surface ozone performance at Zeppelin in terms of R2 and RMSE (Table A1) as compared to other model runs assuming lower values of Tcrit. Since R/V Polarstern (MOSAiC expedition) drifted at locations relatively close to Zeppelin during the spring of 2020, it is probably fair to ask if the model can perform much better using Tcrit = -2.5 deg C for capturing observed surface ozone and BrO variabilities at the MOSAiC ship locations in 2020 (section 4.2). For example, had the surface ozone been more rapidly deleted in the model by the time air masses arrive at the MOSAiC ship locations due to stronger bromine sources arising from the use of Tcrit = -2.5 deg C, simulated BrO concentrations could have been lower at the ship locations and agreed better with measurements from the ship (because the photochemical steady state between Br and BrO under sunlight calls for the presence of ozone to maintain BrO). Relationships among surface bromine source strengths, surface ozone and BrO concentrations are more complex than linear responses because of feedbacks such as those indicated above (e.g., Zhao et al., 2016, section 3.4.2; Hausman and Platt, 1994). *This possibility should be discussed.* Please also read my minor comment on Lines 257-259 concerning the likelihood of bromine explosion occurrences at temperatures as high as close to -2.5 deg C. Nevertheless, the authors have raised valid points about the limitation of the Toyota scheme. I do not claim that simplicity in the Toyota scheme is adequate for very precisely capturing the physical and chemical processes pertaining to the behavior of bromine across the air-snowpack-seaice-seawater system.

Our experiment with $T_{crit}=-2.5$ °C (sfa012) would, most likely, have performed better in May at the location of MOSAiC compared to sfa017. To demonstrate this, we plot a MOSAiC mockup ozone volume mixing ratio (VMR) timeseries evaluated at lat = 85.1°N and lon=2.812°E for Spring 2019 (Fig. 1 (a)). The differences in March/April are small. In May, the difference reaches about 20-25 ppb – but **a complete ozone depletion over a longer time is not reached in sfa012 either**. As suggested by $R^2$ and RSME at Zeppelin, ozone VMR would improve, but mainly in May (Fig. 1 (b)). In consequence, ozone VMR would be far too low at Eureka throughout May and even in June (Fig. 1 (c)). In addition, Fig. 7 (a) in our manuscript indicates that the lower $T_{crit}$, the higher the chance for *false positive ODEs* in Fall (e.g. in late September 2019) which we wanted to avoid.

This said, we could have optimized the parameters for the MOSAiC expedition (or Mt. Zeppelin) in 2020, but our study was not designed for that. In our present study, we explicitly did not look at the data in 2020 both at coastal stations and along the MOSAiC track before settling on the setup to avoid an overfitting.

Regarding the non-linearity of ozone chemistry, we compare sfa012 and sfa017 in terms of ozone and $Br_y$ VMR at Eureka and Mt. Zeppelin in spring 2019 (Appendix Figs. A1-A2). Because we do not reach a complete ozone depletion that could terminate the BrO production, BrO VMR scales with ozone VMR in our experiments. We rephrase L330-331

accordingly following your suggestions. And cite Zhao et al. (2016) and Hausmann and Platt (1994) appropriately in Section 1.

*The photochemical steady state between Br and BrO requires the presence of $O_3$ to maintain BrO production. As the ozone depletion on pan-Arctic scale remains incomplete in our model experiments, the enhanced $\chi_{BrO}$ is a direct consequence.*

[Figure]

Figure 1: Timeseries of ozone mixing ratios in spring 2019 comparing sfa017 and sfa012. Evaluated at (a) lat = 85.1°N and lon=2.812°E (MOSAiC mockup); (b) Zeppelin; (c) Eureka.

2. Figure annotations can and should be made more reader friendly. Table 2 lists all the model experiment numbers/IDs with specific differences among them, but figures and their captions sometimes lack information from which model experiment(s) the presented graphics have been generated. For example, I would add the model experiment IDs (sfa010, sfa011 and sfa012?) on the lefthand side of maps in Figure 6 and/or in the figure caption. Similarly, I believe the authors can add model experiment IDs in Figure 5 or its figure caption. I would also add the model experiment IDs on the lefthand side of scatter plots in Figure A4. Fonts may need to be larger, if possible, to make the annotations legible especially when printed on paper. I had difficulty reading some annotations in Figures 5, 6, A1, A3, and A4 (well, this last one was the most problematic for me; I needed to magnify, magnify and magnify the PDF to be able to read the annotations at last).

We increased the font size of labels, tick labels, and annotations where possible and added the experiment ID either to the caption or as sub-caption of the respective figures listed.

3. Lines 133-136: This paragraph may want to be expanded and detailed a little more. A sentence that briefly describes the relationship between dry deposition velocity and surface resistance can be helpful. More importantly, there is no explicit description for the value of surface resistance assumed on ice/snow surfaces for ozone in the present model runs, but it is discussed later as an

important factor for the simulated behavior of surface ozone by referring to what sounds like a different value (10000 s/m) used/tested in the previous ECHAM/MESSy study (Lines 279-280 and Lines 407-410). Please add a statement in this section on the value of surface resistance for ozone on ice and snow surfaces assumed in the present study.

We dismissed the explicit discussion of dry deposition during the internal iterations of the manuscript, causing an inconsistency in the later section . We include a brief description of the connection between dry deposition velocity ($v^{dd}$), surface resistance ($r_c$), aerodynamical ($r_a$) and quasi-laminar ($r_b$) resistance in the Wesley scheme (Eq. (1)) and the computation of $r_c$ for any species $i$ (Eq. (2)):

$$v^{dd} = \frac{1}{r_a + r_b + r_c}, \tag{1}$$

$$r^i_{c,\,\text{snow}} = \frac{1}{\frac{H_i}{r^{SO_2}_{c,\,\text{snow}} \cdot 10^5} + \frac{R^i_{\text{dry}}}{r^{O_3}_{c,\,\text{snow}}}}, \tag{2}$$

with $r_{c,\,\text{snow}}{}^{O3}$=2000 sm$^{-1}$ and $r_{c,\,\text{snow}}{}^{SO4}$=1 sm$^{-1}$ as suggested.

4. Line 178 and Figure 4: It will be nice if the authors can demonstrate this statement visually by a scatter plot for temperatures vs. their modelled biases with dots colored by BL height.

As suggested, we plot the temperature difference between EMAC and MOSAiC observations (Fig. 1) and include it as subfigure. Observed BL heights are indicated by color. We also show median and mean temperature bias. This figure clearly demonstrates that EMAC is on average between 0-5 °C warmer and most – but not all – data with BL height below 100 m are located both above ΔT=0 °C and the mean/median bias.

[Figure]

Figure 2: Surface temperature difference between EMAC (reference simulation) nudged with ERA5 and MOSAiC observation. The boundary layer heights indicated by color are taken from MOSAiC observation.

5. Lines 280-281: Is it worthwhile referring to the possibility of the Arctic ozone hole in 2020 for having created distinct lower tropospheric photochemical activities in March? The Toyota scheme as implemented in several models to date is unlikely to capture the influence of increased UV irradiance on bromine emissions from the ice surface. Also, although only remotely related to the present study, Steinbrecht et al. (2020) noted a minor impact attributable to 2020 stratospheric ozone depletions in the decrease of ozone from the free troposphere in the CAMS reanalysis data (in which the boundary-layer bromine explosion chemistry was likely neglected). For the model performance weakness in May 2020, I wonder again if Tcrit = -2.5 deg C (not carried out for the year 2020 model run) could have solved part of the problem.

Any effect of low stratospheric ozone (higher UV transmittance) is probably captured in our simulation, as we include the full stratosphere with chemistry in our experiments. Steinbrecht et al. (2021) pointed out that "springtime ozone depletion in the Arctic stratosphere contributed less than one-quarter of the observed tropospheric anomaly". In

their study, they focused on the free troposphere (1-8 km) and the whole (northern) hemisphere. Any direct signal in surface ozone in the Central Arctic would likely be small. The reduced hemispheric background ozone in 2020 (7%) that we do not correctly represent given the recycled pre-COVID ozone precursor emissions, should show the larger impact. As discussed above, $T_{crit}$=-2.5 °C could have reduced $\chi_{O_3}$ in the Central Arctic by about 20-25 ppb compared to sfa017, but would not have reproduced the observed complete depletion of ozone (Fig. (a)).

**[Minor comments]**

Line 36: Was Coburn et al. (2016) a study on Hg chemistry in the polar boundary layer? If not, I suggest citing other references. Brooks et al. (2006) may be a good fit here.

    Coburn et al. (2016) refers more generally to the deposition and formation of toxic, bio-digestible Hg in reaction with BrO over the US. Brooks et al. May indeed be the better fit in the context of this sentence. We substitute the citation accordingly.

Line 48: Magnesium, calcium and potassium are not necessarily negligible cations in natural seawater and are potentially more important than sodium for the maintenance of liquid brines on salty ice surfaces at very low temperatures (e.g., Koop et al., 2000). I suggest the authors revise the statement here from "sodium bromide (NaBr)" to "bromide (Br-)".

    We change the sentence accordingly.

Line 61: Consider citing Adams et al. (2002) and Fickert et al. (1999) in addition to Oldridge and Abbatt (2011). However, the Oldridge and Abbatt study did not really focus on reactions (R6-R8) but examined other physicochemical aspects of bromide oxidation on salty ice surfaces such as the role of acidity and temperature-dependent bromine formation using ozone as an oxidant. In other words, the Oldridge and Abbatt study should be cited more in the context of uncertainties in the process-level understanding. You might want to consider citing Wren et al. (2010) as well along this line.

    We rephrase accordingly.

*This mechanism is well understood through lab experiments (Fickert et al., 1999; Adams et al., 2002) and physicochemical aspects of bromide oxidation on salty ice surfaces including the role of acidity and temperature have been studied in detail (Wren et al., 2010; Oldridge and Abbatt, 2011).*

Lines 61-62: While Sander et al. (2006) used a box model in their study of reactions mentioned here, Toyota et al. (2014) used a one-dimensional model in their study. Aqueous-phase radical reactions rather than Reactions (R6-R8) were suggested to be also important in the deeper snow layers.

    Thank you for the suggestions. We rephrase accordingly:

*Box modeling (Sander et al., 2006) and one-dimensional model studies that also includes deeper snow layers (Toyota et al., 2014) have shown that the picture is far from complete and that liquid-phase reactions also play an important role in deeper snow layers.*

Line 72: Can you be more specific about what was indicated as a source of bromide in Ridley et al. (2003).

    We specify:

*[...] which is in contrast to earlier findings that showed no coincidence between these sea ice features and the intensity of ozone depletion (Ridley et al., 2003).*

Line 96: It will be helpful to indicate the height scales in meters represented by 1-3 lowest model levels.

> The terrain-following hybrid pressure levels in the Arctic translate on average to heights above ground at the upper boundary of each level as follows: level: (1, 2, 3, …, 8) height: (99, 254, 538, …, 3571) m. We include this information in the respective place.

Lines 99-100: "Heterogeneous reactions are mainly restricted to the stratosphere" – does it mean that the heterogeneous reactions in the model are mainly for simulating those taking place in the stratosphere, or in the troposphere? This is an important point that must be clarified because heterogenous reactions on aerosols are among the key factors that control the activity of reactive bromine chemistry in the troposphere including in the polar boundary layer (Fan and Jacob, 1992).

> Heterogeneous reaction rates are computed only in the stratosphere and for polar stratospheric clouds. The respective reactions in the troposphere are, in principle, available as multi-phase reactions for the chemical mechanism, but can currently not be activated in global simulations due to numerical and computational constraints. We therefore parameterized the emission of $Br_2$ from a sea salt aerosol climatology as described in L49-50 and L100-101. We rephrase 99-100:

*The explicit calculation of heterogeneous reactions is restricted to the stratosphere and uses a prescribed aerosol surface concentration climatology […]. In the troposphere and boundary layer, we parameterize the emission of $Br_2$ as described in the following paragraph.*

> In this context, we carefully checked our model setup for the sea salt mass fluxes again and found a mistake in our description of the computation of the sea salt emission flux in L100-101. Instead of the climatology we used the LSCE scheme (Guelle et al., 2001) that computes the mass flux depending on wind speed using lookup tables. We change the sentence accordingly:

*The sea salt aerosol mass flux is computed with the LSCE scheme (Guelle et al., 2001) that applies lookup tables depending on wind speed. A detailed comparison of all available sea salt aerosol emission schemes is found in Kerkweg (2005).*

Line 153: Do you interpolate the model output vertically as well? It could matter for Mt. Zeppelin (474 m ASL), which perhaps resides above the lowest model level, disconnected often from immediate influences from surface emissions near the station both in the real environment and in the model.

> The lowermost level interpolated to the location of Mt. Zeppelin tops at 340 m above mean sea level (center at 230 m). We did not interpolate the model output vertically to the height of the monitoring station. It could be interesting to look more closely into the variance of ozone and other substances with height at this and other locations in the future. For the timeseries of the atmospheric column above the Mt Zeppelin, we find, in general, a weakening of the ODE signal with height and a stronger influence of the background ozone (Fig. 3). So, it could make a difference in cases of elevated BL height.

[Figure]

Fig. 3. Vertical slice of ozone concentrations at Mt. Zeppelin from sfa017.

Lines 205-206: The lack of data coverage by ICDC AoSI automatically should result in the assignment of sea ice age to be less than 1 year in the Canadian archipelago region. Can this be a source of artifact?

The archipelago is covered in both multi-year and first-year sea ice. The assumption that first year sea ice due to its higher salinity contributes most to BEs, potentially introduces a bias towards higher $Br_2$ emissions from that region in our simulations. The BrO VCD above the Canadian archipelago, one of the hotspot regions for BEs, is rather well reproduced by our model experiments. This further underlines that the importance of the age of sea ice is overstated in the Toyota mechanism. Though, it acts as a good proxy for some of the observed patterns.

Lines 209-210: It will be informative if you can explicitly state the range of values given by Regan et al. (2023).

We add:

*The total area of multi-year sea ice based on ICDC AoSI amounts on average to 2.5 $10^6\,km^2$ in the 2020s, which is consistent with the range (0.5-2.9) $10^6\,km^2$ given by Regan et al. (2023) for the time period 2009-2019.*

Table 2 and Lines 255-256: The meaning of "frozen lakes masked" in Table 2 is unclear, so is the meaning of "these are excluded" on Lines 255-256. Do they mean that frozen lakes were neglected from air-surface chemical interactions in corresponding model runs?

Yes, we excluded frozen lakes as reactive surfaces in the respective experiments because they were originally included in our models sea ice concentration maps. This led to false bromine explosions because they were treated like first-year sea ice. The frozen lake mask is used to treat them instead as land snow in the Toyota scheme. We make this clearer in the respective paragraph (below L191):

*Frozen freshwater lakes were originally included in the model's SIC, which caused false BEs around the Great Lakes as these were treated like first-year sea ice. We therefore, first excluded these areas by masking them out in the SIC and later treated them like land snow.*

And in the Table caption:

*Frozen lakes are included in the model's SIC and were either masked out or treated like land snow.*

Line 257-259: Gong et al. (2025) assumed Tcrit close to -2.5 deg C in their model (GEM-MACH) at 15-km grid resolution and found a reasonable surface ozone performance during the spring of 2015 at Zeppelin. As such, we may also argue that the improved surface ozone performance by using Tcrit = -2.5 deg C in ECHAM/MESSy is not necessarily an artifact from inadequate model resolutions and that bromine explosions near Zeppelin may indeed persist at relatively high temperatures as compared to other ice-covered regions for unidentified reasons.

> Thank you for pointing out this study! Indeed, GEM-MACH displays a similar behavior regarding the strength of ODEs at, for example, Mt. Zeppelin compared to Villum – while with Tcri=-2.5°C observation and model align well at Zeppelin, ozone VMR is underestimated at Villum. Including a snow melt factor probably fixed some of the timing issues. We adapt our manuscript accordingly taking these results into consideration:

*An experiment with Tcrit=-2.5°C performs well at Zeppelin, but displayed lower modeled $\chi_{O3}$ than observed at Eureka and Villum research station Nord in May and June (Appendix Fig. B2).*
*At higher spatial resolution (15x15 km), Gong et al. (2025) have shown that, when using an ODE terminator defined by snowmelt and additional bromine emission from multi-year sea ice, observed and modeled $O_3$ VMR at Zeppelin agree well with $T_{crit}$=-2.5 °C, but was underestimated at Villum research station. This suggests that BEs near Zeppelin may persist at relatively high temperatures as compared to other ice-covered regions for unidentified reasons.*

Line 278 (and Figure 7): I am confused about this statement. In Figure 7c, I do see "… in March … a pronounced dip in ozone at Summit" from model time series for March 2020, not from observed time series. In other words, contrary to the authors' statement, I see false decrease in simulated surface ozone from March 2020 not observed at Summit. Here is a side note from me out of this matter: the choice of colors for time series from different model runs and observations could be revised in Figure 7 to make the observed time series more distinct from the model time series.

> The model, indeed, consistently under predicts ozone mixing ratios during winter and even displays spikes towards lower and higher ozone mixing rations at times. We described a feature of dipping ozone concentration occurring in March 2019, coincidentally observed at all station (except maybe for Zeppelin). This feature could be interpreted as ODE but might also hint towards atmospheric dynamics. We should have made named the year in sentence and adapt the text accordingly.

*An ODE in March 2019 at Eureka station […]*

> We also improve the colors of the plots to make model and observational data more distinguishable.

Lines 279-280 and Lines 409-410: Helmig et al. (2007) was the first modelling study in which essentially the same conclusion was reached. Cite this study here.

> We actually followed the work of Helmig et al. (2007) regarding the used surface resistance value in Falk and Sinnhuber (2018) and rephrase accordingly:

*Following Helmig et al (2007), Falk and Sinnhuber (2018) have shown that a higher surface resistance of ozone over ice and snow ($r_c$=10000 sm$^{-1}$) could improve this.*

And for the latter occurrence:

*Following Helmig et al (2007), Falk and Sinnhuber (2018) had shown an improved agreement of ozone VMR between model (MESSy v2.52) […]*

Line 284 and Line 378: Consider citing Steinbrecht et al. (2020) and Weber et al. (2020) as well here.

We added the suggested citations.

Line 285-290: Do oil sector emissions from Prudhoe Bay play a role here as well in the model overprediction of surface ozone in the summer at Utqiagvik?

Thanks for pointing that out. In principle, all places within the probable range of atmospheric transport could cause spikes in modeled ozone. First, one would need to check whether the predominant wind directions or back-trajectories during these events would point back to Prudhoe Bay. Because the modeled emissions in 2019 and 2020 were the same, atmospheric conditions potentially favored the production of ozone in 2020 compared to 2019. But this would be a detailed study of it own.

Lines 329-330: I do not see visual information related to the statement "tropospheric BrO VCD remains lower than observed" at least in Figure 9. Can this information be added to Figure 9?

Here, the reference to Fig. 10(b,d) is missing. We add the reference accordingly.

Line 334: How high is the albedo assumed on sea ice for computing J values in the model runs? If it indeed seems to be too low as an assumption, the authors may want to cite some studies to support their speculation on the snow/ice surface albedo. I also wonder if solar zenith angles as large as 100 degrees (or even greater?) matter for tropospheric twilight chemistry. Or do the authors imply chemistry in the dark? Are there references to back up this part of speculation, too? The range of solar zenith angles indicated here obviously contains typos (100 cannot be smaller than 80), which obscure what is being speculated here.

Thanks a lot for pointing out the typo.

- Regarding the albedo: This speculation ins mainly based on Carns et al. (2015) who measured albedo on sea ice at the coast of Antarctica and the references therein. They refer to lab experiments by Perovich and Grenfell (1981) with artificial (fast grown), pure NaCl ice that had shown an increase in ice albedo with decreasing air temperature peaking at 0.9-1 in the visible range at -37°C. The observations by Carns et al. (2015) indicated the existence of a thin, highly saline snow crust forming on "bare sea ice" with a high albedo of 0.8 in the visible range. On bar sea ice they found a temperature dependent albedo peaking between ~0.7-0.76 in the visible wavelength range.
  In our model, snow covered sea ice albedo is interpolated between a minimum value at $T_{surf} \geq 0°C$ (0.65) and a maximum value at $T \leq -1°C$ (0.8) and for bare sea ice between 0.55 to 0.75. The ocean albedo is set to 0.07. A snow depth of 1 cm is regarded as snow covered, where as the observed saline snow crust had a thickness of 3 mm. From the standard output streams of the model, we currently cannot deduce the effective albedo and in which cases the sea ice was snow covered or bare.
  The radiation submodel uses a grid box average albedo only which reduces the albedo "seen" by the radiation submodel at the sea ice edge and in coastal regions. Whether or not this effective albedo has a significant effect on the photolysis rates would need some more investigation.

- Regarding the solar zenith angle: We consider these solar zenith angles as twilight. Lary and Pyle (1991) have shown the importance of multi-scattering for twilight photolysis (θ>88°) in the stratosphere at a relatively low surface albedo (0.3). In our radiation submodel, j-values are not computed for grid cells, in which the solar zenith angle is greater than 87.5°. Hence, twilight photochemistry may be underestimated unless accounted for elsewhere. This would mainly matter in early spring at the onset of ODEs.

We rephrase and adapt the paragraph accordingly and move it to Section 5.

*Photochemistry could be underestimated in twilight conditions above surfaces with high albedos as we only compute j-values for $\theta_\odot \leq 87°$. For a relatively low surface albedo of 0.3, Lary and McQuaid (1991) had shown that multi-scattering at $80° \leq \theta_\odot \leq 100°$ is non-negligible for stratospheric $NO_x$- and $O_3$-chemistry. At Halley station (Antarctica), the ratio of upwelling to downwelling flux for j-values of selected molecules averaged 0.98 under cloudy conditions, regardless of wavelength or solar zenith angle. Under less cloudy conditions, this ratio increased with wavelength when the solar zenith angle exceeded $75°$ (Jones et al., 2008). At the same time, modeled surface albedo could be underestimated over sea ice with little snow cover. In our model, at a snow depth $\geq 1$ cm and $T < -1 °C$ the surface albedo is 0.8 while for bare sea ice it is 0.75. Measurements by Carns et al. (2015) indicated an albedo of $\sim 0.8$ in the visible wavelength range on a thin, $\sim 0.3$ cm, crust of salty snow.*

Line 353-357: Consider citing Fernandez et al. (2024) here.
We add the reference accordingly.

Line 385: I find the statement "an inherent feature of chemistry and advection in the troposphere" a little too fuzzy. Do the authors want to say, "a consequence of observing different phases of non-linear temporal evolutions in the O3 and BrO concentrations in air masses being transported under the influence of chemistry that leads to BEs and ODEs in the troposphere (e.g., Hausmann and Platt, 1994)"?
Thanks a lot for the suggestion! We change the sentence accordingly.

[Technical suggestions]

Line 30: observed TO DROP below
We change accordingly.
Line 49: constantly -> continually
We change accordingly.
Line 79: spring 2019 CONDITIONS
We change accordingly.
Line 81: climatological ozone depletion TRENDS
We change accordingly.
Line 88: Delete a hyphen before "chemical"
We change accordingly.
Line 95: latitude greater than 68 degrees in both hemispheres
We change accordingly.
Figure 1: Toyota et al. (2012) -> Toyota et al. (2011)
We adapted the figure accordingly.
Line 115: Cl to Br MOLAR ratio
We change accordingly.

Line 120: Henry constant -> Henry's law constant
>
> We change accordingly.

Line 123: Has ODS been defined?
>
> It had been, but apparently not anymore. We include *Ozone Depleting Substances (ODSs)* accordingly.

Line 125: HCFCs = hydrochlorofluorocarbons
>
> We change accordingly.

Line 126: I suppose NMHCs are not part of VOCs in your definition. Do VOCs then represent oxygenated VOCs like acetone?
>
> VOCs include only acetylene ($C_2H_2$).
>
> NMHCs include: $C_2H_4$, $C_2H_6$, $C_3H_6$, $C_3H_8$, $NC_4H_{10}$, $CH_3COCH_3$, $CH_3OH$, HCHO, and MEK.
>
> Terpenes, isoprene and monoterpene, are emitted using MEGAN in the onemis submodel.

Line 131: surface concentrations concentrations -> surface concentration climatologies
>
> We change accordingly.

Line 134: dry deposition resistances -> dry deposition velocities
>
> We change accordingly.

Line 136: Henry coefficient -> Henry's law coefficient
>
> We change accordingly.

Table 1 caption: Delete "Referred". Data sources could include more information such as websites and the date of data download in the references.
>
> We refer to the data availability section and include the information about the latest access accordingly.

Line 172: was -> were
>
> We change accordingly.

Line 173: Which data did Mahajan (2022) report?
>
> Mahajan (2022) reported both Br and BrO presented by Benavent et al. (2022). We change the sentence accordingly and update the data availability section.

Line 177: -2 -> 2
>
> We change accordingly.

Line 178: -10 -> 10
>
> We change accordingly.

Line 178: This WARM bias increases with decreasing temperatures
>
> We change accordingly.

Figure 4 caption: Temperature -> surface air temperature; BL -> BL height
>
> We change accordingly.

Line 191: Any citable document for the release candidate of MESSy v2.56?
>
> No, the respective gitlab repository is currently only open for consortium members. A citation of the merge request would therefore not help much.

Table 2: MESSy model version (2.55.2) and MECCA chemistry (CCMI2-base-01) are all the same among model runs. Do they need to be presented like this in the table? Also, what is the difference between "diagnostic" and "interactive" approaches for sea salt Br2? Can it be clarified by a more detailed description in the text? Lastly, a note on sfa002 indicates that this run was used for the CATCH arctic bromine model intercomparison project, which becomes clearer only when we come to read acknowledgements. If the authors wish to indicate this fact about sfa002, it should probably be done more clearly within Table 2 or in Section 3.

- We adapt the table as suggested and move the repetitive information to the table caption.
  *We use MESSy v2.55.2 including two critical bug fixes in the AirSnow mechanism that have been integrated into the MESSy release candidate. The MECCA chemistry mechanism CCMI2-base-01 has been applied for all experiments.*

- Sea salt bromine was not treated as chemical tracer by default, thus we refer to it as "diagnostic". "Interactive" means that sea salt bromine was added to the $Br_2$ chemical tracer in the lowermost level affecting bromine release in the Toyota scheme. We adapt the text: *Sea salt bromine was not treated as chemical tracer by default, here referred to as* diagnostic.
- and table caption accordingly:
  *By default, bromine prescribed from the sea salt aerosol mass flux was not treated as chemical tracer, referred to as* diagnostic *in Table 2. We refer to the inclusion of $Br_2$ to the tendency of the respective chemical tracer at the lowest atmospheric level as* interactive.
- We specify the model intercomparison in Table 2 caption.
  *Experiment sfa002 has been submitted to the* *Arctic Bromine Model Intercomparison project*.

Figure 5 caption: a model experiment -> model experiments
> We change accordingly.

Line 215: Is the "potential vorticity tropopause index" a commonly used terminology? If not, I would rephrase it to a more descriptive statement like "we … used a dynamical tropopause metric based on the model potential vorticity (PV) at X (1.0, 1.6, or 2.0?) Potential Vorticity Unit (1 PVU = 10-6 m2 s-1 K kg-1) to separate …"
> We rephrase accordingly.

Line 224: temporally in springtime -> in time and space during the spring
> We change accordingly.

Line 233: lower threshold -> higher threshold temperature
> We change accordingly.

Line 237: this does not result in NOTABLY enhanced BrO VCD
> We change accordingly.

Lines 237-239: From this sentence, it is not very clear what was measured in a study by Jalkanen and Manninen (1996). Did they measure chloride and bromide concentrations in particulate matter or more of their total gaseous and particulate concentrations? Referring to their sampling and detection methodologies might clear things up.
> This paragraph is indeed not correct in this context and we will remove it in our next revision. Jalkanen and Manninen (1996) sampled aerosols and analyzed them for their bromine, chlorine, sodium, sulfur, nitrogen, and metal content. From the partitioning of these tracers they identified the main sources of the sampled aerosols as mainly marine or anthropogenic of origin – corresponding to the major wind directions during the sampling period. The data indicate a bromide to chlorine enhancement of the marine aerosols in the coarse and fine mode at Utö compared to a ratio of 1/667.

Lines 239 and 245: Tcrit = -2.5
> We change accordingly.

Line 260: AS STATED ABOVE, tropospheric BrO VCD showed a QUALITATIVEBLY better RESULT with
> We change accordingly.

Figure 6 caption: … flux integral for April 2019 …
> We change accordingly.

Line 273: contained -> confined
> We change accordingly.

Line 276: late April/early May -> late April to early May
> We change accordingly.

Line 277: late March/late April -> late March to late April
> We change accordingly.

Line 283: emissions of these -> their emissions

We change accordingly.

We change accordingly.

We change accordingly.

We change accordingly.

We change accordingly.

We change accordingly.

Lines 303-304: Please be more specific about what "this" refers to in this sentence.

We specify:

*The close-to-equal distribution of $O_3$ VMR during the ODE season [...]*

Line 309: atmospheric background O3 has increased -> background O3 concentrations apparently have increased

We change accordingly.

Line 318: an underestimation of the frequency of χO3 becoming greater than 37 ppbV

We change accordingly.

Line 340-343: Consider citing Peterson et al. (2019) and referring to discussions therein in regard to the role of multi-year ice snowpacks as a source of gaseous bromine for the Arctic boundary layer.

We add the following reference to Peterson et al. (2019):

*These findings support the conclusions of Peterson et al. (2019), indicating that regions covered by multi-year sea ice serve as a more significant source of reactive bromine than previously recognized.*

We change accordingly.

We change accordingly.

We change accordingly.

We change accordingly.

We change accordingly.

We change accordingly.

We change accordingly.

We change accordingly.

We change accordingly.

Figure A1 caption: Revise the second sentence to "Comparison between T42 and T106 illustrates the potential benefit of a higher model resolution for resolving inhomogeneity in the characteristics of sea ice such as its age."

We change accordingly.

Figure A2 caption: Add "monthly" before "time series" if that is the case. I do not quite understand the meaning of the second sentence. Please rephrase. The third sentence may sound clearer if you say "MYSIC total areas estimated the two methods differ by 30%".

We remove the second sentence as it does no longer make sense in this context. We rephrase:

*Annual (ERA5) and monthly (ICDC) mean time series of the total area covered by multi-year sea ice in the Arctic. MYSIC total areas estimated the two methods differ by 30 %.*

Figure A3 caption: integral FOR April 2019

We change accordingly.

**Bibliography**

Adams, J. W., Holmes, N. S., and Crowley, J. N.: Uptake and reaction of HOBr on frozen and dry NaCl/NaBr surfaces between 253 and 233 K, Atmos. Chem. Phys., 2, 79–91, doi:10.5194/acp-2-79-2002, 2002.

Benavent, N., Mahajan, A. S., Li, Q., Cuevas, C. A., Schmale, J., Angot, H., Jokinen, T., Quéléver, L. L. J., Blechschmidt, A.-M., Zilker, B., Richter, A., Serna, J. A., Garcia-Nieto, D., Fernandez, R. P., Skov, H., Dumitrascu, A., Simões Pereira, P., Abrahamsson, K., Bucci, S., Duetsch, M., Stohl, A., Beck, I., Laurila, T., Blomquist, B., Howard, D., Archer, S. D., Bariteau, L., Helmig, D., Hueber, J., Jacobi, H.-W., Posman, K., Dada, L., Daellenbach, K. R., and Saiz-Lopez, A.: Substantial contribution of iodine to Arctic ozone destruction, Nat. Geosci., 15, 770–773, https://doi.org/10.1038/s41561-022-01018-w, 2022.

Brooks, S. B., A. Saiz-Lopez, H. Skov, S. E. Lindberg, J. M. C. Plane, and M. E. Goodsite (2006), The mass balance of mercury in the springtime arctic environment, Geophys. Res. Lett., 33, L13812, doi:10.1029/2005GL025525.

Carns, R. C., Brandt, R. E., and Warren, S. G.: Salt precipitation in sea ice and its effect on albedo, with application to Snowball Earth, J. Geophys. Res. Oceans, 120, 7400–7412, https://doi.org/10.1002/2015JC011119, 2015.

Falk, S. and Sinnhuber, B.-M.: Polar boundary layer bromine explosion and ozone depletion events in the chemistry-climate model EMAC v2.52: implementation and evaluation of AirSnow algorithm, Geosci. Model Dev., 11, 1115–1131, https://doi.org/10.5194/gmd-11-1115-2018, 2018.

Fan, SM., Jacob, D. Surface ozone depletion in Arctic spring sustained by bromine reactions on aerosols. Nature 359, 522–524 (1992). https://doi.org/10.1038/359522a0.

Fernandez, R. P., Berná, L., Tomazzeli, O. G., Mahajan, A. S., Li, Q., Kinnison, D. E., Wang, S., Lamarque, J.-F., Tilmes, S., Skov, H., Cuevas, C. A., and Saiz-Lopez, A.: Arctic halogens reduce ozone in the northern mid-latitudes, P. Natl. Acad. Sci. USA, 121, e2401975121, https://doi.org/10.1073/pnas.2401975121, 2024.

Fickert, S., J. W. Adams, and J. N. Crowley (1999), Activation of Br2 and BrCl via uptake of HOBr onto aqueous salt solutions, J. Geophys. Res., 104(D19), 23719–23727, doi:10.1029/1999JD900359.

Gong, W., Beagley, S. R., Toyota, K., Skov, H., Christensen, J. H., Lupu, A., Pendlebury, D., Zhang, J., Im, U., Kanaya, Y., Saiz-Lopez, A., Sommariva, R., Effertz, P., Halfacre, J. W., Jepsen, N., Kivi, R., Koenig, T. K., Müller, K., Nordstrøm, C., Petropavlovskikh, I., Shepson, P. B., Simpson, W. R., Solberg, S., Staebler, R. M., Tarasick, D. W., Van Malderen, R., and Vestenius, M.: Modelling Arctic lower-tropospheric ozone: processes controlling seasonal variations, Atmos. Chem. Phys., 25, 8355–8405, https://doi.org/10.5194/acp-25-8355-2025, 2025.

Grenfell, T. C., and D. K. Perovich (1981), Radiation absorption coefficients of polycrystalline ice from 400–1400 nm, *J. Geophys. Res.*, 86(C8), 7447–7450, doi:10.1029/JC086iC08p07447.

Guelle, W., Schulz, M., Balkanski, Y., and Dentener, F.: Influence of the source formulation on modeling the atmospheric global distribution of sea salt aerosol, J. Geophys. Res. Atmos., 106, 27 509–27 524, https://doi.org/10.1029/2001JD900249, 2001.

Hausmann, M., and U. Platt (1994), Spectroscopic measurement of bromine oxide and ozone in the high Arctic during Polar Sunrise Experiment 1992, J. Geophys. Res., 99(D12), 25399–25413, doi:10.1029/94JD01314.

Helmig, D., Ganzeveld, L., Butler, T., and Oltmans, S. J.: The role of ozone atmosphere-snow gas exchange on polar, boundary-layer tropospheric ozone – a review and sensitivity analysis, Atmos. Chem. Phys., 7, 15–30, https://doi.org/10.5194/acp-7-15-2007, 2007.

Jones, A. E., Wolff, E. W., Salmon, R. A., Bauguitte, S. J.-B., Roscoe, H. K., Anderson, P. S., Ames, D., Clemitshaw, K. C., Fleming, Z. L., Bloss, W. J., Heard, D. E., Lee, J. D., Read, K. A., Hamer, P., Shallcross, D. E., Jackson, A. V., Walker, S. L., Lewis, A. C., Mills, G. P., Plane, J. M. C., Saiz-Lopez, A., Sturges, W. T., and Worton, D. R.: Chemistry of the Antarctic Boundary Layer and the Interface with Snow: an overview of the CHABLIS campaign, Atmos. Chem. Phys., 8, 3789–3803, https://doi.org/10.5194/acp-8-3789-2008, 2008.

Kerkweg, A.: Global Modelling of Atmospheric Halogen Chemistry in the Marine Boundary Layer, Ph.D. thesis, Rheinische Friedrich-Wilhelms-Universität Bonn, https://nbn-resolving.org/urn:nbn:de:hbz:5N-06365, 2005.

Koop, T., A. Kapilashrami, L. T. Molina, and M. J. Molina (2000), Phase transitions of sea-salt/water mixtures at low temperatures: Implications for ozone chemistry in the polar marine boundary layer, J. Geophys. Res., 105(D21), 26393–26402, doi:10.1029/2000JD900413.

Lary, D. and McQuaid, J.: Diffuse radiation, twilight, and photochemistry ? II, J. Atmos. Chem., 13, 373–392, https://doi.org/10.1007/BF00057753, 1991.

Mahajan, A.: Substantial contribution of iodine to Arctic ozone destruction - data, https://doi.org/10.17632/BN7YTZ4MFZ.1, 2022.

Oldridge, N. W. and Abbatt, J. P. D.: Formation of Gas-Phase Bromine from Interaction of Ozone with Frozen and Liquid Na-Cl/NaBr Solutions: Quantitative Separation of Surficial Chemistry from Bulk-Phase Reaction, J. Phys. Chem. A, 115, 2590–2598, https://doi.org/10.1021/jp200074u, 2011.

Peterson, PK, et al. 2019. Snowpack measurements suggest role for multi-year sea ice regions in Arctic atmospheric bromine and chlorine chemistry. Elem Sci Anth, 7: 14. DOI: https://doi.org/10.1525/elementa.352

Steinbrecht, W., Kubistin, D., Plass-Dülmer, C., Davies, J., Tarasick, D. W., von der Gathen, P., et al. (2021). COVID-19 crisis reduces free tropospheric ozone across the Northern Hemisphere. Geophysical Research Letters, 48, e2020GL091987. https://doi.org/10.1029/2020GL091987

Weber, J., Shin, Y. M., Staunton Sykes, J., Archer-Nicholls, S., Abraham, N. L., & Archibald, A. T. (2020). Minimal climate impacts from short-lived climate forcers following emission reductions related to the COVID-19 pandemic. Geophysical Research Letters, 47, e2020GL090326. https://doi.org/10.1029/2020GL090326

Wren, S. N., T. F. Kahan, K. B. Jumaa, and D. J. Donaldson (2010), Spectroscopic studies of the heterogeneous reaction between O3(g) and halides at the surface of frozen salt solutions, J. Geophys. Res., 115, D16309, doi:10.1029/2010JD013929.

Zhao, X., K. Strong, C. Adams, R. Schofield, X. Yang, A. Richter, U. Friess, A.-M. Blechschmidt, and J.-H. Koo (2015), A case study of a transported bromine explosion event in the Canadian high arctic, J. Geophys. Res. Atmos., 120, doi:10.1002/2015JD023711.

**Appendix A**

[Figure]

Figure A1. Modeled (sfa012, sfa017) ozone and bromine species timeseries in 2019 at Eureka.

[Figure]

Figure A2. Modeled (sfa012, sfa017) ozone and bromine species timeseries in 2019 at Zeppelin.

---

## Author Response (AR2)

**Authors response to editor comments (2025-09-26)**

Thank you for your comments helping to improve the accessibility of our article to a general audiance.

**General comments**

- 1. Please consider the Copernicus style.
- Section should be abbreviated as "Sect." unless it appears at the begin of the sentence
- Northern Hemisphere -> northern hemisphere
- Title should be in small letters (not starting each word with a capital letter)

We adapt the respective parts.

2. Figure labels appear at the bottom, but usually these are put at the left or right side of the figures. Thus my question would be if you could adjust the labels to appear at the left or right side of the panels?

We adapt the subfigure labels as requested.

**Specific comments**

- Figure 4 caption: Full stop after last sentence missing (or misplaced? There is a small point one line below)

Thanks for pointing this out. The point was misplaced.

- L239: Don't use dots between units and number We change as suggested.

- Figure 6: Here the figure panes have labels (i), (ii) and (iii) and then additionally (a)-(f). This double labeling is a bit unfortunate and also inconsistent since the left column panels have no additional labels.

We have remove the additional labeling and label each figure separately.

- L249: I would suggest to write "squared Pearson correlation coefficient" instead of "Pearson correlation coefficient squared".

We adapt as suggested.

- L275: consult -> see

We adapt as suggested.

- L279-280: The brackets [] should be replaces by parenthesis ().

We adapt as suggested.

- L318: Add sentence to previous paragraph (to avoid having a paragraph consisting of one sentence).

We adapt as suggested.

- L327, 324 and 355: ppbV -> ppbv. Note, at some places there is only ppb written. We adapt as suggested.

- L356: ppm -> ppmv?

We cannot find any ppm in the given line.  $\rightarrow$  "( $\chi_{03} \le 5$  ppb)"

- L412: put "e.g." before the reference.

We adapt as suggested.

- L421: didn't -> did not

We adapt as suggested.

- L426: ppb -> ppbv?

We adapt as suggested.

- Figure B2: There are no differences visible. Adjust color scale to make these visible?

For early Spring (March to April), no differences are expected. The difference between the simulations only affects late Spring (May) IF surface temperatures rise are above -10 °C. We add this remark to the figure caption.

- Summary and conclusion: It confused me quite a lot that your last sentence of the abstract appeared in the middle of the conclusion. After that the summary and conclusion section read more like a discussion. Thus, I was wondering if it wouldn't be worth to put some part of the summary and conclusion section into a own discussion section?

We revise and separate discussion and summary and conclusions as suggested and add a dedicated section for discussion.